# Co-translational assembly orchestrates competing biogenesis pathways

Maximilian Seidel [1,2], Anja Becker[1], Filipa Pereira[3,4], Jonathan J. M. Landry[5],
Nayara Trevisan Doimo de Azevedo[5], Claudia M. Fusco [6], Eva Kaindl[1], Natalie Romanov[1], Janina Baumbach[1,3],
Julian D. Langer [6,7,8], Erin M. Schuman [6], Kiran Raosaheb Patil[3,9], Gerhard Hummer [10,11],
Vladimir Benes [5] & Martin Beck [1,3✉]

During the co-translational assembly of protein complexes, a fully synthesized subunit engages with the nascent chain of a newly synthesized interaction partner. Such events are thought to contribute to productive assembly, but their exact physiological relevance remains underexplored. Here, we examine structural motifs contained in nucleoporins for their potential to facilitate co-translational assembly. We experimentally test candidate structural motifs and identify several previously unknown co-translational interactions. We demonstrate by selective ribosome profiling that domain invasion motifs of beta-propellers, coiled-coils, and short linear motifs may act as co-translational assembly domains. Such motifs are often contained in proteins that are members of multiple complexes (moonlighters) and engage with closely related paralogs. Surprisingly, moonlighters and paralogs assemble co-translationally in only some but not all of the relevant biogenesis pathways. Our results highlight the regulatory complexity of assembly pathways.

[1] Department of Molecular Sociology, Max Planck Institute of Biophysics, Frankfurt, Germany. [2] Faculty of Bioscience, Heidelberg University, Heidelberg, Germany. [3] Structural and Computational Biology Unit, European Molecular Biology Laboratory (EMBL), Heidelberg, Germany. [4] Life Sciences Institute, University of Michigan, Ann Arbor, MI, USA. [5] Genomics Core Facility, European Molecular Biology Laboratory (EMBL), Heidelberg, Germany. [6] Department of Synaptic Plasticity, Max Planck Institute for Brain Research, Frankfurt, Germany. [7] Membrane Proteomics and Mass Spectrometry, Max Planck Institute of Biophysics, Frankfurt, Germany. [8] Mass Spectrometry, Max Planck Institute for Brain Research, Frankfurt, Germany. [9] Medical Research Council Toxicology Unit, University of Cambridge, Cambridge, United Kingdom. [10] Department of Theoretical Biophysics, Max Planck Institute of Biophysics, Frankfurt, Germany. [11] Institute of Biophysics, Goethe University Frankfurt, Frankfurt, Germany. ✉email: martin.beck@biophys.mpg.de

Protein complexes are a key organizational unit of the proteome. Their modular composition has facilitated the evolution of a very diverse repertoire of folds and corresponding functions. To maintain this very diverse repertoire within the crowded cellular environment poses a logistic burden as the energy gap favoring specific over nonspecific binding decreases with proteome complexity[1]. Therefore, it has been proposed that assembly pathways impose a major restraint on the evolution of protein complexes[2], whereby duplication events of subunits during divergent evolution may necessitate the diversification of protein interfaces or sophisticated quality control mechanisms to avoid promiscuous binding[3]. Co-translational interactions of nascent polypeptides with their respective binding partner have been discovered for many eukaryotic protein complexes[4–7]. Homomeric complexes may assemble by co-co assembly in which either nascent chains emerge from consecutive ribosomes of the same mRNA entangled in *cis*, or alternatively from multiple mRNAs that are clustered by nascent chain interactions in *trans*[5,8]. In contrast, many heteromers may rely on co-post assembly[6,7,9], which we will further refer to as co-translational assembly. Here, a soluble, fully synthesized subunit binds to the nascent polypeptide chain of the interactor. Such co-translational assembly events contribute to orphan protein stability and solubility and may be coordinated with the association of assembly chaperones[4]. It has been proposed that they may be beneficial for nascent chain folding or non-promiscuous stoichiometric assembly[9]. It has been hypothesized that they seed assembly pathways when moonlighting interactions are possible[10]. Since co-translational assembly pathways of moonlighters remain largely unexplored, the exact physiological contribution of distinguished co- and/or post-translational assembly pathways remains uncertain.

Nuclear pore complexes (NPC) perforate the nuclear envelope (NE) to facilitate nucleocytoplasmic exchange. They are among the largest, non-polymeric, eukaryotic assemblies and are composed of ~30 different nucleoporins (Nups) that constitute a multi-layered modular architecture of astonishing complexity[11,12]. Beyond canonical protein interfaces of nucleoporin subcomplexes of up to 10 components, various other types of interactions are crucial for the formation of the higher ordered, eightfold rotational symmetric structure of ~500 nucleoporins in yeast[12,13]. Those include weak interactions of intrinsically disordered Phenylalanine-Glycine (FG)-rich repeats contained in so-called FG-Nups that function as a velcro[14] and short linear motifs (SLiMs) within so-called linker Nups that facilitate interactions within and across subcomplexes[15,16]. Furthermore, structured motifs such as beta-propeller complementation[17–19] and coiled-coil interactions[16,20,21] are observed in multiple instances. Interestingly, different Nup subcomplexes that have evolved from each other may contain shared or closely related subunits that assemble promiscuously in vitro[22–25]. Nevertheless, it remains unclear how such promiscuous interactions are suppressed or discriminated in vivo.

Due to the importance of the NPC as a permeability barrier, faithful assembly imposes a challenge for cells, which is addressed by different pathways depending on the spatiotemporal context[13]. While NPCs are made from pre-existing building blocks during post-mitotic assembly in higher eukaryotes, they are synthesized from scratch during the ubiquitous interphase assembly pathway[13] and *Drosophila* oogenesis[26,27]. Interphase assembly is the only known biogenesis pathway in yeast and spatially proceeds from the inside-out at the NE[28]. Although the rough order of subcomplex recruitment to membranes has been resolved[13,28–30], little is known about the early steps of assembly that may occur away and independently from membranes. Besides local[27] and some co-translational events[31] that have been discovered during NPC assembly, it remains unclear which of the above-introduced motifs are subject to such events, in which order they intertwine into the assembly pathways, and how exactly they contribute to faithful assembly.

Here, we elucidate the role of a subset of such motifs for the co-translational de novo assembly of NPC subcomplexes during interphase assembly in *Saccharomyces cerevisiae*. We show that these events orchestrate complex formation in competitive scenarios.

## Results

**A framework to investigate co-translational assembly of Nups.** The competition between the specific interactions stabilizing a complex and the far more numerous non-specific promiscuous interactions intensifies with increasing numbers of complex subunits and possible interaction partners[1]. This imposes a severe challenge for the biogenesis of very large complexes with many subunits such as the NPC (Fig. 1a). Co-translational assembly can tolerate higher levels of such competitive binders because it increases the dwell-time of synthesis intermediates at the ribosome in which surfaces that potentially engage in promiscuous interactions are not yet exposed. One would thus predict that cells harness the power of co-translational assembly for the biosynthesis of NPC subcomplexes in the cytosol. Here, we extend our previous theory[1] by including co-translational assembly as a possible assembly enhancer. We derived a mathematical model that captures the essence of this process (Supplementary Note 1), namely how failures in the assembly at intermediate steps accumulate and how this impacts the overall success (Supplementary Fig. 1a, b). This model further reveals that the overall yield of protein complex assembly pathways decreases exponentially with the number of subunits (Supplementary Fig. 1c). It indicates several ways in which co-translational assembly can increase the yield, in particular by capturing intermediates that would otherwise be prone to misassembly or aggregation. In addition, co-translational assembly introduces a hierarchy into the process that is effectively cutting down the number of steps. For small assemblies, this may be an insignificant gain. However, for large assemblies such as the NPC, hierarchical assembly should significantly increase the overall success. Co-translational assembly may also increase the on-rate of a newly synthesized component to a partial complex retained in the vicinity of the ribosome, such that the rate of assembly will be limited by the translation speed instead of the slow diffusion of low abundance species.

We surveyed the known structural repertoire of nucleoporins for domains that could potentially engage in co-translational interactions because they (i) are small interaction motifs found in linker Nups; (ii) complement the fold of another nucleoporin; or (iii) are shared between multiple complexes and thus could be promiscuous interactors (Fig. 1b). This concerns either the competitive binding of Nups for the same binding domains within the NPC or the integration of Nups into distant, functionally unrelated complexes. To elucidate how co-translational association of Nups contributes to the faithful assembly of the NPC, we experimentally validated these motifs in a hypothesis-driven approach.

We first generated a library of C-terminally Twin-StrepII tagged[32] Nups using a scar-free cloning technique in *S. cerevisiae*[33]. Scar-free cloning preserves the endogenous 3′ untranslated region (3′UTR) of a messenger RNA and avoids changes that may affect mRNA fate and translation[34]. We used these strains for affinity purification of the respective StrepII-tagged baits (Supplementary Fig. 2a) and analyzed the co-enriched mRNAs by quantitative real-time PCR adapting previously established methods[4,6,7]. Below we refer to this

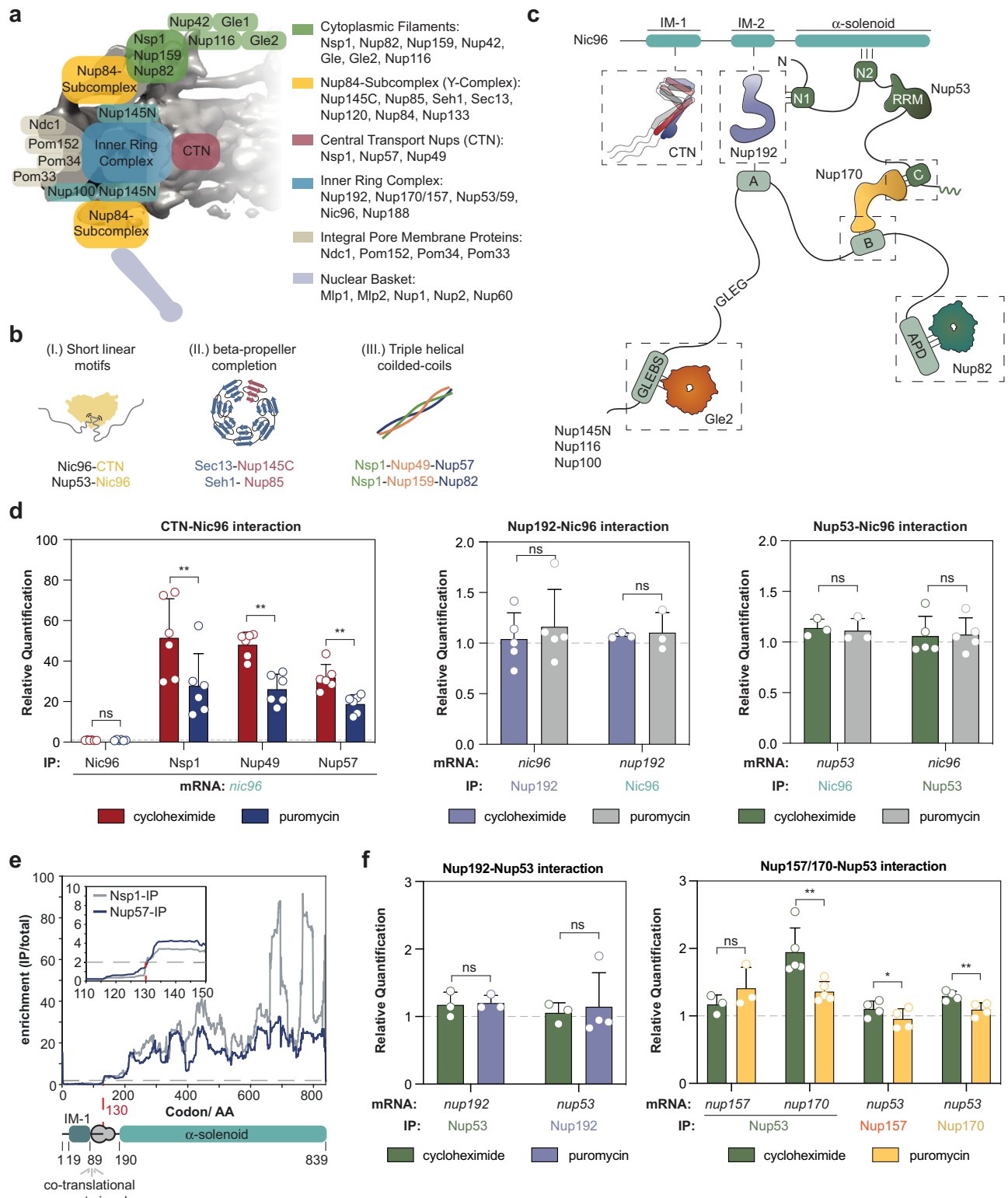

method as RIP-qPCR (Supplementary Fig. 2b). As a positive control, we reproduced the known co-translational interaction of fatty acid synthase subunit Fas1 with nascent Fas2[4], represented by enrichment of *fas2*-mRNA, which was above our signal threshold of 1.5. This interaction was sensitive to the translation-specific inhibitor puromycin, which causes dissociation of the nascent chain and thereby enables the dissection of co-translational interactions from cryptic RNA binding activity[35].

Inverse tagging of Fas2 instead of Fas1 did not enrich for either mRNA, as expected (Supplementary Fig. 2c–e).

**Sequential assembly of central transport Nups.** Using the above standards for the experimental validation, we tested the potential co-translational interactions of various full-length Nups that engage with linker Nups. First, we investigated the co-translational landscape of Nic96 which contains several motifs

**Fig. 1 Co-translational assembly of linker Nups. a** Scheme of the NPC architecture. Subcomplexes are mapped to the *in cellulo* structure of the *S. cerevisiae* NPC (EMD: 10198)[45]. **b** Recurring structural features of the NPC: (I) short linear motifs and small structured domains, (II) incomplete beta-propellers in the Nup84-subcomplex and (III) triple helical coiled-coils. **c** Scheme of previously determined interaction motifs contained in linker Nups that may help to organize the assembly in vivo. Adapted from Beck and Hurt (2017)[11]. **d** RIP-qPCR experiments with affinity purifications of CTN/Nic96 (*left*) that associate via the IM-1, Nup192/Nic96 (*middle*) that interact via the IM-2 and Nup53/Nic96 that interact via N2 (*right*), imply that the entire CTN binds co-translationally to Nic96. $n = 6$ biologically independent samples for Nsp1-StrepII (*nic96*-mRNA), Nup49-StrepII (*nic96*-mRNA), Nup57-StrepII (*nic96*-mRNA), Nic96-StrepII (*nic96*-mRNA); $n = 5$ biologically independent samples for Nup192-StepII (*nic96*-mRNA) and Nup53-StepII (*nic96*-mRNA) and $n = 3$ biologically independent samples for Nic96-StepII (*nup192*- and *nup53*-mRNA). $**p = 0.0078$ for Nsp1-StrepII (*nic96*-mRNA); $**p = 0.0041$ for Nup49-StepII (*nic96*-mRNA) and $p** = 0.0026$ for Nup57-StepII (*nic96*-mRNA). **e** SeRP experiments with affinity purifications of Nsp1 and Nup57 reveal a synchronous co-translational binding within the *nic96*-transcript. Data was derived from $n = 4$ biologically independent samples. **f** RIP-qPCR for Nup192 and Nup53 (*left*) that interact via the N1 motif and for Nup157/170 and Nup53 (*right*) that interact using the C-motif. $n = 5$ biologically independent samples for Nup53-StepII (*nup170*-mRNA); $n = 4$ biologically independent samples for Nup192-StepII (*nup53*[puromycin]-mRNA), Nup157-StepII (*nup53*-mRNA) and Nup170-StepII (*nup53*-mRNA) and $n = 3$ biologically independent samples for Nup192-StepII (*nup53*[cycloheximide]-mRNA), Nup53-StepII (*nup192*- and *nup157*-mRNA). $**p = 0.0078$ for Nup53-StepII (*nup170*-mRNA); $*p = 0.0127$ for Nup157-StepII (*nup53*-mRNA) and $**p = 0.0075$ for Nup170-StepII (*nup53*-mRNA). Bar graphs in panel **d**, and **f**, depict mean ± SD. ns $p > 0.05$, $*p < 0.05$, $**p < 0.01$ (Two-sided, paired t-test). Source data for RIP-qPCR in panel **d**, and **f**, are provided as a Source Data file. AA amino acid, IP immunoprecipitation.

to connect the inner ring to the nuclear envelope and the central transport Nup (CTN) trimer (Fig. 1c)[15,16,36]. Our RIP-qPCR data indicate a co-translational interaction of all three components of the CTN with the nascent chain of Nic96 (Fig. 1d). We did not observe inverse mRNA enrichment for any subunit of the CTN in Nic96 RIP-qPCR experiments (Supplementary Fig. 3a) or co-translational interactions with Nup192 or Nup53 (Fig. 1d).

To investigate if the nascent IM-1 domain of Nic96 co-translationally binds to Nsp1 and Nup57, we turned to selective ribosome profiling (SeRP), as previously described[4] (Supplementary Figs. 4 and 5a). This method relies on the comparison of ribosome-protected mRNA footprints from a total translatome to those that are affinity-purified using a specific bait protein that may engage in interactions with nascent chains. Thereby the onset of the co-translational interaction within a given open reading frame (ORF) is revealed. As a positive control, we reproduced the co-translational onset of the interaction of full-length Fas1 with the nascent chain of Fas2. We observed an enrichment of footprints within the *fas2*-transcript precisely at the previously reported onset[4] at the Fas2 assembly domain (Supplementary Figs. 5b, c, 6).

The selective ribosome profiles generated from Nsp1- and Nup57-IPs for *nic96*-mRNA show simultaneous binding properties to the nascent chain at codon 130. Considering a 30–40 amino acid offset of nascent chain accessibility due to the ribosome exit tunnel[37], the measured onset coincides with the exposure of the IM-1 motif of Nic96 from the exit tunnel (Fig. 1e). The synchronous onset further underscores the notion that only fully assembled CTN trimer associates with Nic96[15,20,38]. SeRP analysis further indicated that Nic96 does not engage with nascent chains of the other known interactors (Supplementary Figs. 6 and 7a).

Independent of Nic96, Nup53 and Nup192 can also directly associate with one another via an N1 interaction site of Nup53[39] (Fig. 1c). However, we did not detect co-translational assembly of Nup53 with Nup192 or vice versa (Fig. 1f). Nup53 further binds to Nup170/157 with the so-called C-motif[36]. Interactions with both, Nup157 and Nup170 were reported in vitro[40], but integrative modelling highlights the physiological relevance of the interaction with Nup170[21,41]. We found that Nup53 co-translationally binds to nascent Nup170 but not Nup157 (Fig. 1f). In our RIP-qPCR experiment, Nup170 and Nup157 did display significant signal reduction for *nup53*-mRNA upon puromycin treatment. However, the small effect size (<1.5) and additional SeRP experiments (Supplementary Fig. 6) do not support that this reduction was due to co-translational engagement. It is interesting

to note that co-translational assembly events are detected only on the physiologically relevant pathway.

**Differential co-translational assembly events govern paralogous assembly pathways.** We next examined the paralogous linker proteins, Nup100, Nup116, and Nup145N. Interactions of the three linker Nups with Nup192 and Nup170/157 are mediated by the A- and B-motifs[36] (Fig. 1c). The N-terminal GLEBS domain of Nup116 binds to Gle2 but is absent in Nup100 or Nup145N[42,43] (Fig. 2a). Indeed, we found that Gle2 binds co-translationally to the nascent chain of Nup116 but not Nup145N or Nup100 (Fig. 2b). We did not detect any additional co-translational events using Nup53, Nup157, Nup170, and Nup192 as baits (Fig. 2c, d, Supplementary Fig. 3b). The interaction of Nup157 with nascent Nup145N was only slightly below the threshold of 1.5-fold enrichment. Validation using SeRP did not detect any ribosome footprint enrichment for *nup145*-mRNA in a Nup157-SeRP experiment (Supplementary Fig. 6).

The C-terminal autoproteolytic domains (APD) of Nup145N, Nup100 and Nup116 bind to the cytoplasmic filament protein Nup82 in vitro[15,44]. However, integrative[21] and in situ structural analysis[45] stresses the physiological relevance of the interaction with Nup116. Our RIP-qPCR data reveal that Gle2, Nup116 and Nup82 form a hierarchical co-translational assembly chain (Fig. 2e, Supplementary Fig. 3c) that may determine Nup116-specificity for the cytoplasmic filaments. Taken together, our data indicate that some but not all motifs contained in linker Nups engage in co-translational interactions. These co-translational association events may help to specify the paralogous assembly pathways (Fig. 2f).

**Domain invasion motifs can act as co-translational assembly domains.** Inspired by the above findings, we wondered whether co-translational events may also specify assembly pathways for moonlighters that are members of multiple protein complexes. Two members of the Nup84-subcomplex, Seh1 and Sec13, are incomplete beta-propellers lacking one blade, that interact with the WD40 domain invasion motifs (DIM) of Nup85 and Nup145C, respectively[18,46]. Seh1 and Sec13 both share moonlighting functions in the Seh1-associated (Sea) complex (Supplementary Fig. 7b)[47,48]. We first investigated the domain invasion of Nup85 into the incomplete beta-propeller of Seh1 (Fig. 3a)[18]. RIP-qPCR analysis revealed that Seh1 enriched for the *nup85*-mRNA, but not the *sea4*-mRNA, in a translation-dependent manner (Fig. 3b). To address if the co-translational

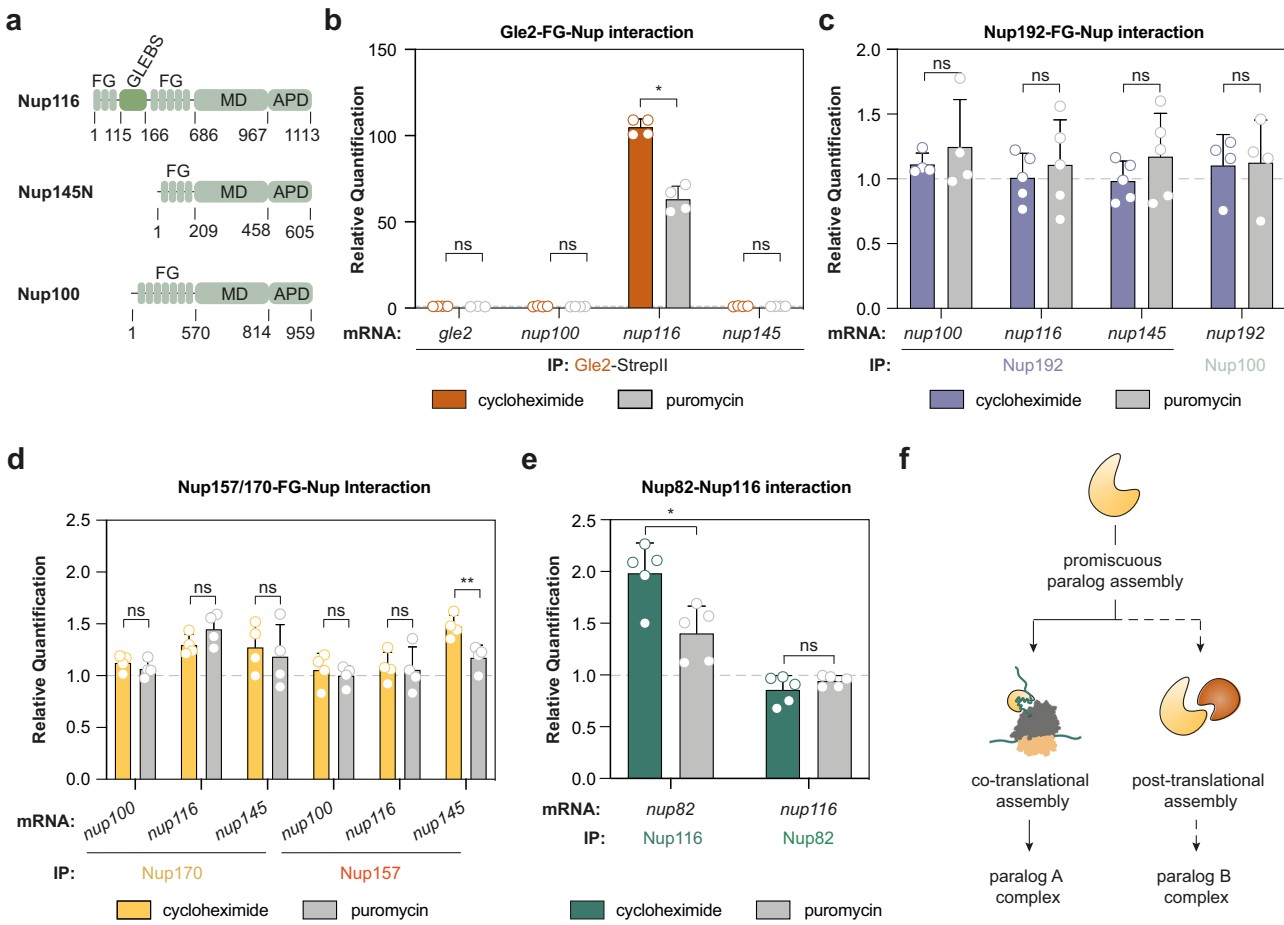

**Fig. 2 Paralogues linker Nups co-translationally engage with their binding partners. a** Primary structure scheme of the paralogous FG-Nups Nup145N, Nup116 and Nup100. **b** RIP-qPCR experiments with affinity purified Gle2 that show co-translational binding to nascent Nup116. $n = 4$ biologically independent samples for Gle2-StepII (*gle2*-, *nup100*-, *nup116*-, and *nup145*-mRNA). *$p = 0.0143$ for Gle2-StepII (*nup116*-mRNA). RIP-qPCR was used to characterize co-translational interactions between **c**, Nup192 and Nup100 **d**, Nup157 and Nup170 with linker Nups and **e** Nup116 and Nup82. $n = 5$ biologically independent samples for Nup192-StepII (*nup116*- and *nup145*-mRNA), Nup116-StepII (*nup82*-mRNA) and Nup82-StepII (*nup116*-mRNA); n = 4 biologically independent samples for Nup192-StepII (*nup100*-mRNA), Nup100-StepII (*nup192*-mRNA), Nup157-StepII (*nup100*-, *nup116*- and *nup145*-mRNA) and Nup170-StepII (*nup100*-, *nup116*- and *nup145*-mRNA). **$p = 0.0023$ for Nup157-StepII (*nup145*-mRNA) and *$p = 0.0296$ for Nup116-StepII (*nup82*-mRNA). **f** Scheme highlighting that designated assembly pathways may exist for paralogs. Bar graphs in panel **b–e** depict mean ± SD. ns $p > 0.05$,* $p < 0.05$, ** $p < 0.01$ (Two-sided, paired *t*-test). Source data for RIP-qPCR in panel **b–e**, are provided as a Source Data file. AA amino acid, IP immunoprecipitation, FG phenylalanine-glycine repeats, MD middle domain, APD autoproteolytic domain.

association of full-length Seh1 with the nascent chain of Nup85 is indeed mediated by the domain invasion motif at its N-terminus, we turned to SeRP. In line with our RIP-qPCR experiments, SeRP of Seh1-IPs identified an enrichment of footprints within the *nup85*-ORF (Fig. 3c). We analyzed the translatome-wide data but did not identify similarly strong enrichment patterns of footprints for any other gene (Supplementary Figs. 6 and 7c–f). This analysis suggests that the members of the Sea-complex, which are known to interact with Seh1[47,48] but that remain to be structurally analyzed at high resolution, do not mimic assembly intermediates of Nup85. Surprisingly, the onset of the co-translational interaction of Seh1 with nascent Nup85 does not coincide with the emergence of the domain invasion motif (residue 44–101) from the exit channel of the ribosomes. It rather maps to alpha-helices and their connecting loops at positions 405–544. Although distant in sequence, these helices are located right beneath the domain invasion motif in the structure of the Seh1-Nup85 heterodimer[49]. One may speculate that similar to the Sec13-Sec31 heterodimer[50], interactions of Seh1 with the helical bundle may restrict the flexibility of the alpha-solenoid (Fig. 3d) (see discussion).

Sec13 is part of the COPII coatomer complex[51,52], the nuclear pore[17,46] and the Sea-complex[47,48] where it binds the domain invasion motifs of at least three of the four known interactors. To investigate the role of Sec13 during the assembly of the respective complexes, we purified Sec13, Sec31, Nup145C, and the Sea-complex protein Mtc5 and analyzed the co-eluted mRNAs by qPCR (Fig. 4a). We found that Sec31 can co-translationally engage with its own mRNA. The IP against Sec13 enriched for *sec31*-mRNA; however, puromycin treatment only weakly perturbed this interaction. We do not find any other purified component enriched for *sec13*-mRNA.

We further analyzed the co-translational interaction network of Sec13 and Sec31 using SeRP (Fig. 4b). The resulting data were in line with our targeted RIP-qPCR approach. Sec13 showed an onset at codon 437 within the *sec31*-ORF at the position of the DIM (Fig. 4b, c). In contrast, the interaction of Sec31 with nascent Sec31 shows an onset that is shifted to the dimerizing interface at codon 680 (Fig. 4b, d). Sec13 furthermore engages co-translationally with Sec16. This onset is shifted downstream of the DIM to helices within the alpha-solenoid that are located beneath the beta-propeller prior to codon 1367 (Fig. 4b, e), similar

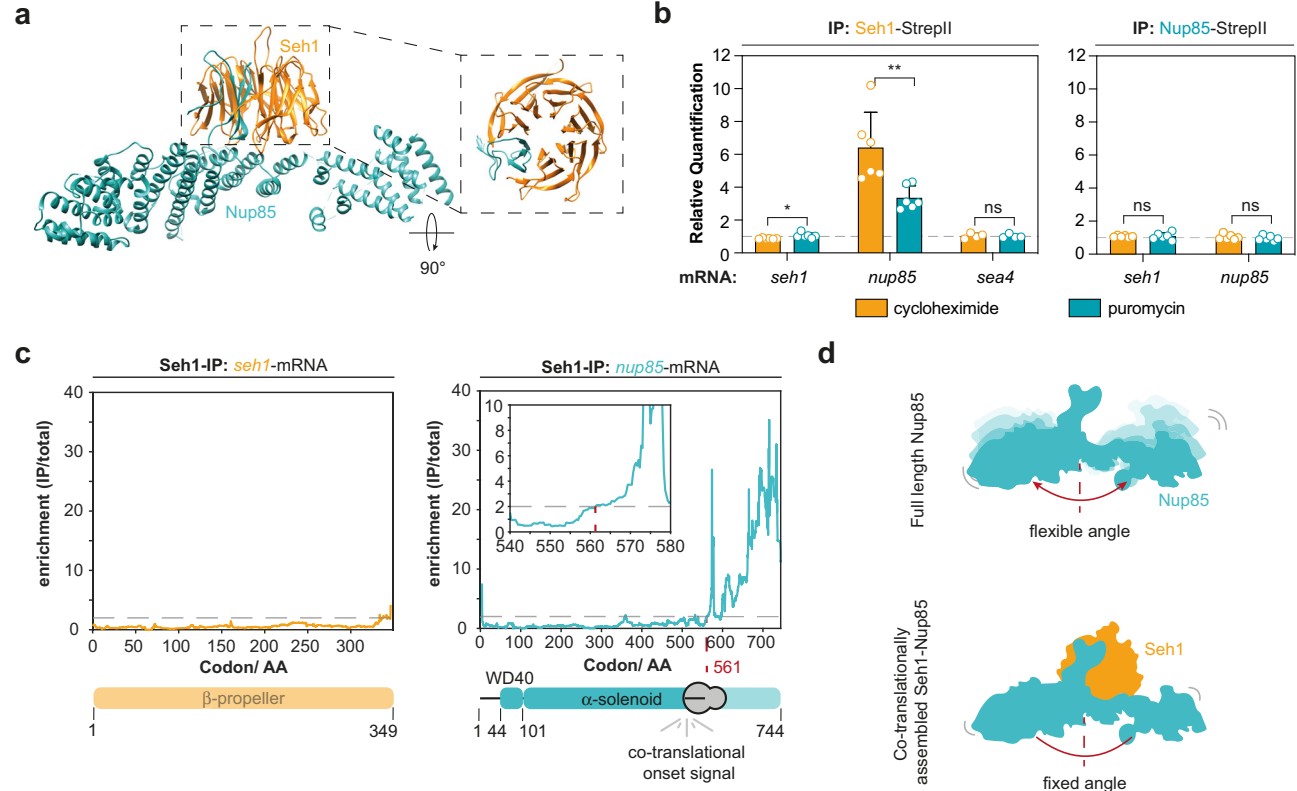

**Fig. 3 Beta-propellers can be complemented co-translationally in vivo. a** Crystal structure of Seh1 bound to Nup85 (PDB: 4XMM)[49]. **b** RIP-qPCR reveals co-translational entanglement of Seh1 with Nup85. Bar graphs show mean ± SD. $n = 6$ biologically independent samples for Seh1-StepII (*seh1*- and *nup85*-mRNA) and Nup85-StepII (*seh1*- and *nup85*-mRNA) and $n = 4$ biologically independent samples for Seh1-StepII (*sea4*-mRNA). *$p = 0.0178$ for Seh1-StepII (*seh1*-mRNA) and **$p = 0.0065$ for Seh1-StepII (*nup85*-mRNA). **c** Selective ribosome profiling data derived from a Seh1-IP shows that helices at the trunk of Nup85 but not the domain invasion motif (WD40-domain), are required for stable association with Seh1. Selective ribosome profiling was performed with $n = 4$ biologically independent replicates. **d** Seh1 might restrict the flexibility of the alpha-solenoid by binding to a structural joint represented by the helices it attaches to. ns $p > 0.05$, *$p < 0.05$, ** $p < 0.01$ (Two-sided, paired t-test). Source data for RIP-qPCR in panel **b**, are provided as a Source Data file. AA amino acid, IP immunoprecipitation, NPC nuclear pore complex.

to the mode of binding observed in Seh1-Nup85 (Fig. 3c, d). The superimposed structures of the insertion blades of Sec16 and Sec31 show only subtle differences in their fold (Fig. 4f). Surprisingly, the known interaction partner within the NPC, Nup145C, did only show very weak signal and no clearly defined onset (Supplementary Fig. 7g). Taken together, these findings illustrate that interaction motifs may co-translationally engage with some but not necessarily all of their interaction partners and that the exact onset of co-translational interactions may adapt in a versatile way to protein folding.

**Coiled coils of the NPC assemble co-translationally.** The CTN-subcomplex is densely packed to the equatorial plane of the central channel of the inner ring. It consists of the Nsp1, Nup57, and Nup49 proteins. They contain N-terminal FG-rich intrinsically disordered domains that interact with nuclear transport receptors[53]. Each of the three members further contains a coiled-coil domain that hetero-trimerizes thus forming the scaffold of the subcomplex (Fig. 5a). To elucidate if and how co-translational assembly events within the CTN-subcomplex contribute to the organization of the assembly pathway, we first studied the co-translational landscape by RIP-qPCR. Our results suggest that Nsp1 co-translationally binds to Nup57, but not Nup49 (Fig. 5b). This is in line with previous in vitro analysis showing that Nsp1 cannot directly recruit Nup49 and hence Nup57 acts as an organizer subunit of the trimeric coiled-coil[38]. To specify the onset of co-translational entanglement of Nsp1 with nascent

Nup57 we used SeRP (Fig. 5c). These experiments highlighted that the minimal requirement for co-translational interactions is the initial coiled-coil segment 1 (CCS1) that forms a rather long rod-shaped stretch. Neither RIP-qPCR nor selective ribosome profiling experiments detected a translation-dependent interaction of Nsp1 or Nup57 with the *nup49*-mRNA (Fig. 5b, c and Supplementary Figs. 6 and 7h) suggesting that Nup49 is added to the complex post-translationally.

Intriguingly, the evolutionary related cytoplasmic filament subcomplex that functions as a platform for mRNA export[54], is predicted to form a similar hetero-trimeric arrangement as the aforementioned CTN trimer (Fig. 5d)[21,55]. It consists of Nup159, Nup82, and yet again Nsp1, whereby Nup57 and Nup82 compete for the same binding site within the Nsp1 coiled-coil region[22]. We wondered if the cytoplasmic filaments are constructed by a similar assembly pathway. Surprisingly, RIP-qPCR and SeRP experiments targeting Nup82 and Nup159 showed no signal (Fig. 5e, f, compare to 5b, 5c). However, we found that Nup159 co-translationally engages with nascent Nup82 (Fig. 5e), suggesting that Nsp1 is rather added post-translationally to the subcomplex. The inverse order of Nsp1 incorporation that we observed for both subcomplexes may help to specify their composition (Fig. 5g).

**Perturbation of co-translational assembly in vivo.** Although the assembly pathways for CTN- and CF-subcomplexes are distinct, the domain architecture of the C-terminal coiled-coil segments of

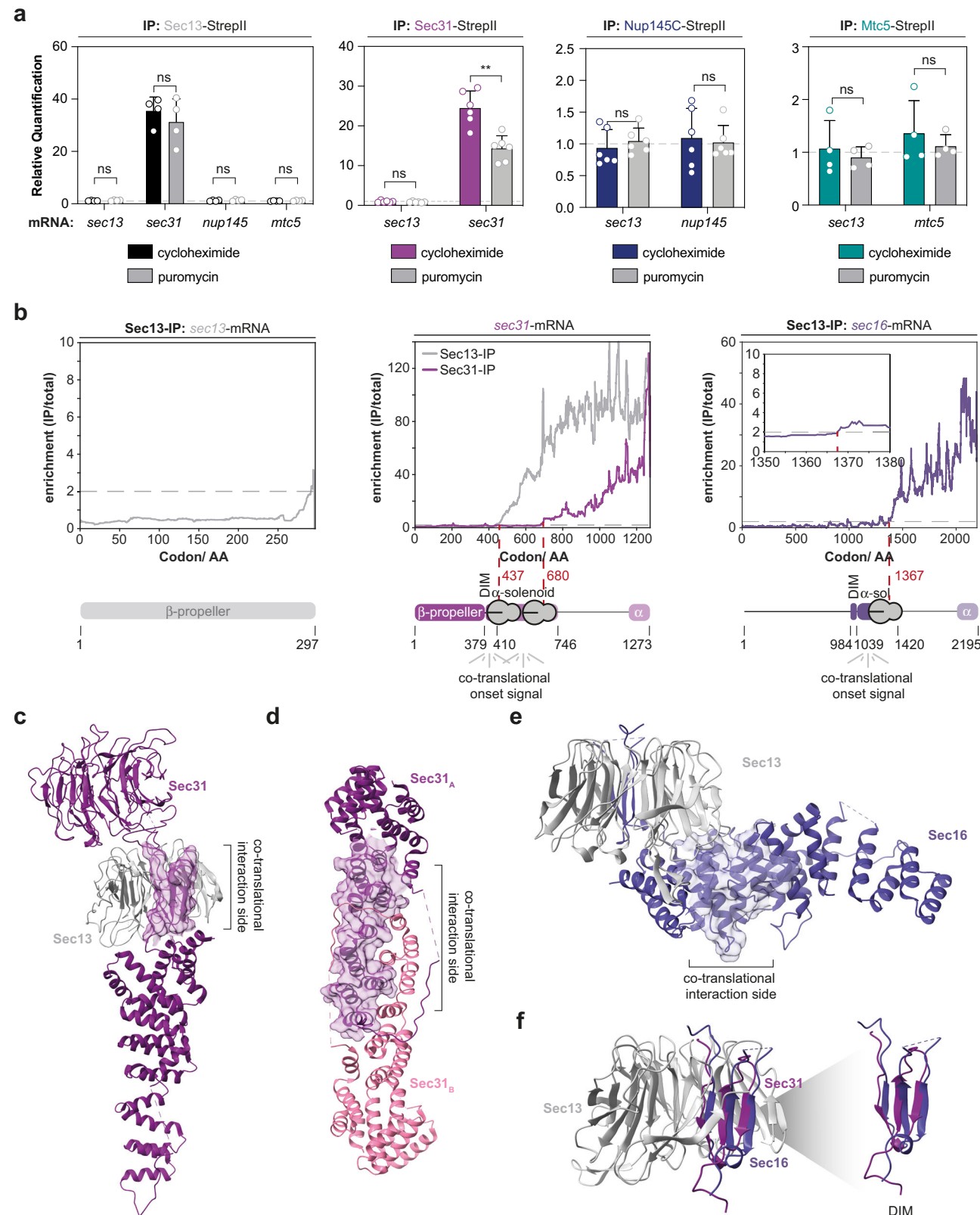

Nup57 and Nup82 are structurally similar (Fig. 6a). A notable exception is a small alpha-beta domain unique to Nup57; upstream of the CCS1 that is expanded to a ferredoxin-like domain in vertebrates[16,20]. Deletion of this domain in Nup57 led to non-stoichiometric CTN-subcomplexes during in vitro reconstitution[16]. We set out to test if the alpha-beta domain is required for co-

translational interactions of Nsp1 with nascent Nup57 in vivo. We deleted the alpha-beta domain by scar-free cloning and further designed two chimeric mutants in which we substituted the CCS1 of Nup57, with either the entire CCS1 of Nup82 thereby removing the alpha-beta domain, or a truncated CCS1 of Nup82 thereby maintaining the alpha-beta domain (Fig. 6a). Removal of the

**Fig. 4 Co-translational substrate recognition is modular. a** RIP-qPCR experiments with Sec13, Sec31, Nup145C, and Mtc5 as baits. Bar graphs depict mean ± SD. $n = 6$ biologically independent samples for Sec31-StepII (*sec13-* and *sec31*-mRNA) and Nup145C-StepII (*sec13-* and *nup145*-mRNA) and $n = 4$ biologically independent samples for Sec13-StepII (*sec13-*, *sec31-*, *nup145-* and *mtc5*-mRNA) and Mtc5-StepII (*sec13-* and *mtc5*-mRNA). **p = 0.004 for Sec31-StepII (*sec31*-mRNA). **b** SeRP analysis of Sec13 and Sec31 from $n = 3$ biologically independent replicates. **c** Sec13 recognizes the domain invasion motif of Sec31 (highlighted as transparent isosurface) in a co-translational manner **d**, Sec31-Sec31 dimerization occurs after release of the entire alpha-solenoid interaction surface (highlighted as transparent isosurface) from the exit channel of the ribosome. Structures in **c**, and **d**, are taken from 4BZK[62]. **e** The co-translational interaction of Sec13 with nascent Sec16 occurs subsequent to the synthesis of alpha-helices located below the beta-propeller of Sec13 that are highlighted as transparent isosurface (PDB: 3MZK[51]). **f**, Superposition of the domain invasion motifs of Sec16 and Sec31 within Sec13 (PDB: 4BZK, 3MZK). ns $p > 0.05$, *$p < 0.05$, **$p < 0.01$ (Two-sided, paired *t*-test). Source data for RIP-qPCR in panel **a**, are provided as a Source Data file. AA amino acid, IP immunoprecipitation, DIM domain invasion motif.

alpha-beta domain did not impact thermo-sensitivity of the strain implying that overall fitness remained unaffected (Fig. 6b). Contrarily, the CCS1 chimeric mutants show mild growth defects upon the removal of a plasmid-encoded *nup57* rescue copy suggesting a fundamental role of CCS1 in faithful CTN formation (Fig. 6b). We used quantitative mass spectrometry to asses complex composition in Nsp1 pull-downs from Nup57 wildtype and mutant strains. CTN components were mildly increased in binding to Nsp1 in the Nup57 alpha-beta deletion strain, while the Nup159-subcomplex remained largely unaffected. In contrast, the CCS1 chimeric mutants effectively decreased the fraction of co-enriched CTN components indicating compromised assembly (Fig. 6c and Supplementary Figs. 8a, b). Although alpha-beta-domain deletion mildly perturbed the integrity of the CTN, it did not impair the co-translational assembly of Nsp1 - Nup57 or CTN - Nic96 (Fig. 6d). In contrast, CCS1 chimeric mutants resulted in the loss of co-translational enrichment of *nup57-* and *nic96*-mRNA in a Nsp1 RIP-qPCR experiment. Surprisingly, the chimeric mutants display a previously undetected co-translational enrichment for *nup82*-mRNA (Fig. 6d). The reason why the Nup82 CCS1 is not sufficient for binding of Nsp1 when duplicated in the Nup57-ORF might be explained by the SeRP data that were obtained under wildtype conditions (Fig. 5f). It showed a slight enrichment that was C-terminally shifted with respect to *nup57*-ORF and located subsequent to the CCS3 of Nup82, which was not included in the Nup57 chimeric mutant. We thus speculate that the Nsp1 interaction with nascent Nup82 would normally be suppressed due to higher affinity of Nsp1 to Nup57 CCS1 but becomes relevant once effective binding to Nup57 is impaired (Fig. 5g and Supplementary Fig. 8c).

## Discussion

Our hypothesis-driven approach has identified many co-translational assembly events that are relevant throughout all stages of NPC biogenesis with local hotspots at the subcomplexes utilizing Nsp1 (Fig. 6e). Our qPCR experiments are in line with many, though not all interaction pairs reported by a recent screen[31]. Specifically, we identified translation-dependent interactions of Seh1—*nup85*-mRNA, CTN—*nic96* mRNA, and Nsp1—*nup57*-mRNA, while we obtained negative data for Nup192 - *nup100*-mRNA and Nup157—*nup145*-mRNA. Differences may be attributed to the experimental design, in particular the use of translation-specific inhibitors such as cycloheximide and puromycin, the biochemical conditions, the design of scarlessly cloned yeast strains and SeRP analysis.

We found that moonlighting proteins that are part of multiple complexes may co-translationally engage in some but not necessarily all alternative assembly pathways. This is exemplified by Seh1 that does co-translationally interact with Nup85 but not with the Sea-complex, Sec13 that co-translationally interacts with Sec31 and Sec16 but not Nup145C or Mtc5, as well as Nsp1 that does co-translationally interact with Nup57 but not with Nup82, unless binding to Nup57 is perturbed. Remarkably, the

paralogous Nup159- and CTN-subcomplexes contain the most co-translational interactions. These data suggest that co-translational assembly may be used to organize assembly pathways in a hierarchical manner when multiple outcomes are possible. An interesting aspect is that the domain invasion motifs of Nup85 and Sec16 are not sufficient for the co-translational interactions with Seh1 and Sec13, respectively, but require the trunk of the supporting alpha-helical domains. One may speculate that Seh1 rigidifies the alpha-solenoid of Nup85 (Fig. 3d). This could be beneficial (i) to promote the interaction with Nup120[49,56] and/or (ii) to induce membrane curvature at the nuclear envelope. This is exemplified by the Sec13-Sec31 heterodimer[50]. Here, an increase of rigidity was associated with the induction of membrane curvature of COPII vesicles and therefore might be reminiscent to the inside-out extrusion of the nuclear envelope in interphase assembly[57].

Sec13 and Seh1 also exemplify that the onset of co-translational interactions may adapt in a versatile way to protein folding. In fact, it appears likely that the domain invasion motifs are subject to stronger selection pressure because its binding interfaces in the incomplete beta-propeller are the same for each interactor and therefore only allow for subtle changes. Consequently, other structural features may be evolutionarily more accessible to organize a unique assembly pathway for the interactors of moonlighting proteins. The fact that minor changes in the open binding interfaces in the incomplete beta-propeller of Seh1 and Sec13 contribute to the selection of different binding partners was previously demonstrated by attempts to substitute Sec13 with Seh1 in a Sec31-Seh1 fusion construct within *sec13Δ-S. cerevisiae* strains. The fusion was lethal, highlighting the idea that the domain invasion blades are tailor-made for their native interactors[50].

In our study, the CTN-subcomplex nicely emphasizes how co-translational association may orchestrate assembly pathways if promiscuous interactions could form. The physiological CTN composition was controversial and had been addressed using different techniques[16,20,23–25]. Our data point to a model where Nsp1 binds the CCS1 domain of nascent Nup57 in a co-translational manner, which is crucial for preventing the formation of the promiscuous Nup49:Nup57:Nup49 trimer[23]. Full length Nup49 is then bound in a post-translational manner. The resulting heterotrimer binds to the IM-1 assembly domain of Nic96, yet again co-translationally. This experimentally determined assembly outline agrees with our theoretical considerations (Supplementary Note 1) that suggest a benefit from the hierarchal organization (Supplementary Fig. 1). It shows that not necessarily all, but at least several of the individual steps of a given assembly pathway may occur co-translationally (Supplementary Fig. 8c). Our chimeric Nup57 mutants interfere with the respective co-translational interaction with Nsp1 in vivo and highlight its benefits. In this case, the perturbation of a co-translational complex biogenesis reduced the efficiency of complex formation (Supplementary Fig. 8b) as well as fitness (Fig. 6b). It resulted in

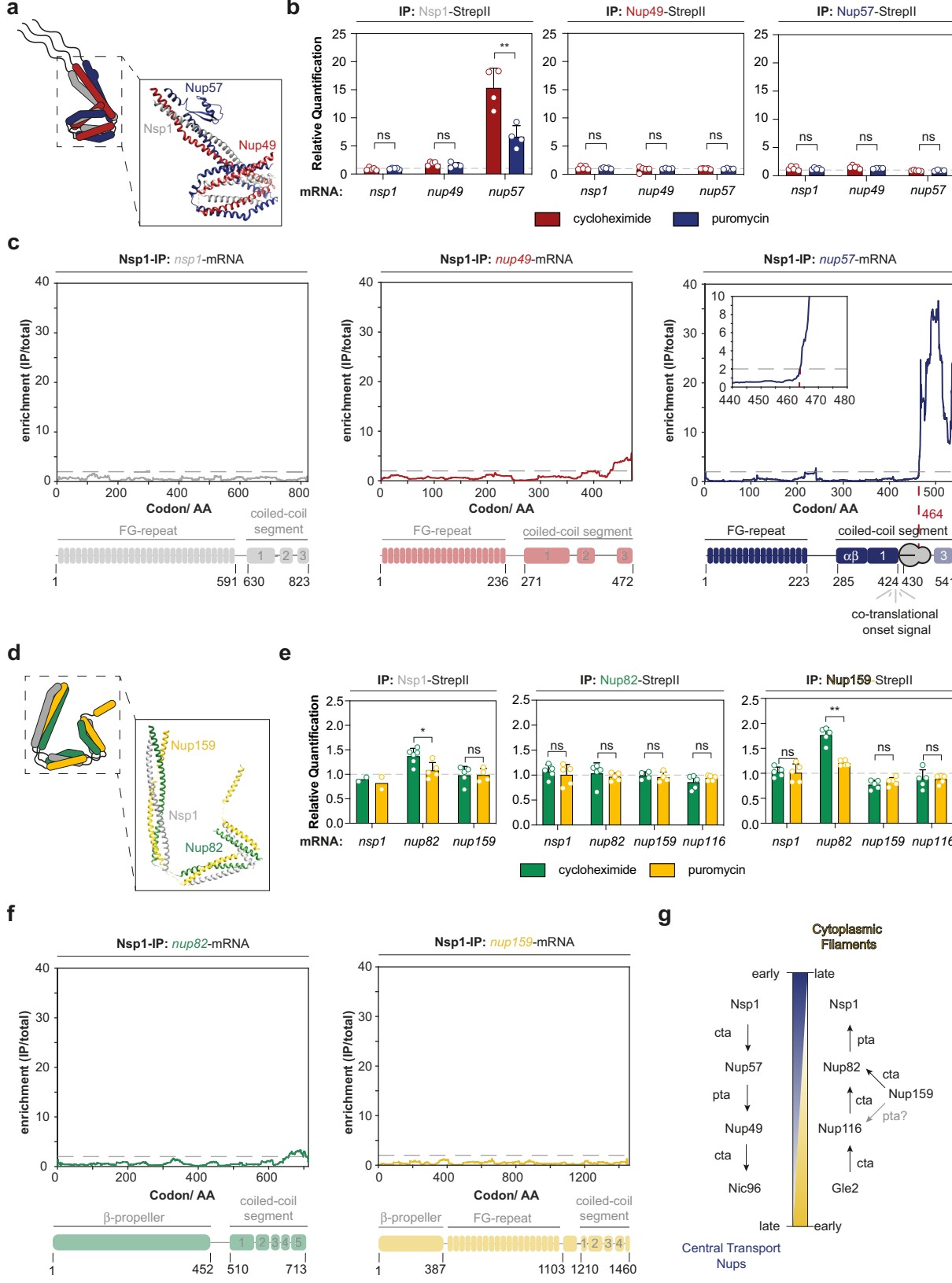

competitive assembly scenarios (Fig. 6d). The consequent co-translational engagement of Nsp1 with nascent Nup82 interfered with the native linear outline of the respective assembly pathway (Supplementary Fig. 8c).

Taken together with the recent mass spectrometric study that has revealed the temporal order of subunit engagement[29], our findings provide new insights into NPC biogenesis. It has been previously shown that most of the Nup-subcomplex encoding mRNAs, including those of the CTN complex (Nup62 in *Drosophila melanogaster*) are dispersed in the cytosol[27]. The advantage of translation under relatively dilute conditions in the cytosol could be that stoichiometric subcomplexes are formed but higher-

**Fig. 5 Nsp1 engages with the CTN and cytoplasmic filament subcomplexes in two opposing assembly pathways. a** Structure of *Chaetomium thermophilum* CTN shows the heterotrimeric coiled-coil which is tethered to Nic96 (PDB: 5CWS)[16]. **b** RIP-qPCR of the CTN suggests co-translational interactions of Nsp1 with nascent Nup57 but not Nup49. Bar graphs show mean ± SD. $n = 5$ biologically independent samples for Nsp1-StepII (*nsp1*- and *nup49*-mRNA), Nup49-StepII (*nsp1*[cycloheximide]-, *nup49*[cycloheximide]- and *nup57*-mRNA) and Nup57-StepII (*nsp1*-, *nup49*-, and *nup57*-mRNA) and $n = 4$ biologically independent samples for Nsp1-StepII (*nup57*-mRNA) and Nup49-StepII (*nsp1*- and *nup49*-mRNA under puromycin). **$p = 0.0042$ for Nsp1-StepII (*nup57*-mRNA). **c** Selective ribosome profiling from Nsp1-IPs identifies its co-translational association with nascent Nup57 at the coiled-coil segment 1 (CCS1). Selective ribosome profiling data was generated from $n = 4$ biologically independent samples. **d** Structural model of the coiled-coil in the Nup159-subcomplex[21]. **e** RIP-qPCR experiments targeting the Nup159-subcomplex (affinity purified Nsp1, Nup82 and Nup159). Nup159 co-translationally binds to nascent Nup82. RIP-qPCR experiments depicted as mean ± SD. $n = 6$ biologically independent samples for Nsp1-StepII (*nup82*-mRNA under cycloheximide); $n = 5$ biologically independent samples for Nsp1-StepII (*nup82*[puromycin]- and *nup159*[cycloheximide]-mRNA), Nup82-StepII (*nsp1*-, *nup82*- and *nup116*-mRNA) and Nup159-StepII (*nsp1*-, *nup82*- and *nup116*-mRNA), $n = 4$ biologically independent samples for Nsp1-StepII (*nup159*-mRNA under puromycin) and Nup82-StepII (*nup159*-mRNA) and $n = 2$ biologically independent samples for Nsp1-StepII (*nsp1*-mRNA). *$p = 0.0465$ for Nsp1-StepII (*nup82*-mRNA) and **$p = 0.0015$ for Nup159(*nup82*-mRNA). **f** SeRP with affinity purified Nsp1 does not detect co-translational association within the Nup159-subcomplex. Selective ribosome profiling was performed with four biologically independent replicates. **g** Assembly scheme for the CTN- and Nup159-subcomplexes. Nsp1 co-translationally seeds the assembly in the CTN-subcomplex but post-translationally completes the assembly of the Nup159-subcomplex. ns $p > 0.05$, *$p < 0.05$, **$p < 0.01$ (Two-sided, paired t-test). Source data for RIP-qPCR in panel **b**, and **e**, are provided as a Source Data file. IP immunoprecipitation, AA amino acid, FG phenylalanine-glycine repeats, cta co-translational assembly, pta: post-translational assembly.

order interactions, such as subcomplex oligomerization, are prevented due to the low protein concentrations. The benefit of co-translational assembly could be to increase the dwell-time of assembly intermediates to nevertheless facilitate efficient assembly. Hereby, at least four early modules are synthesized separately, the Nup84-, CTN-, Nup159-, and inner ring complexes. Next, subcomplexes are recruited to sites of NPC biogenesis in proximity to membranes where their local concentration is increased and higher-order interactions across subcomplexes, such as ring formation, become kinetically favored. Here, Nup53 and Nup170 that were previously assembled co-translationally seed the recruitment of further components such as Nup188 and Nup192[29] presumably already associated with Nup100[31]. Subsequently, additional subcomplexes are recruited, various of which were pre-assembled stoichiometrically in a co-translational manner. The concept of co-translational assembly thus elegantly complements present scientific models of NPC biogenesis and explains how promiscuous interactions are avoided in such a very complex macromolecular assembly.

## Methods

**Yeast strains and growth media**. StrepII-tagged yeast strains were obtained by scar-free homologous recombination using the MX4blaster cassette[33]. Briefly, the MX4blaster cassette was amplified with gene-specific overhangs for homologous recombination. PCR products were transformed and positive clones were selected on YPD-high phosphate plates supplemented with 300 µg/mL hygromycin B (ForMedium) and 3 g/l potassium dihydrophosphate (monobasic). To remove the MX4blaster cassette, MX4 positive clones were grown in low phosphate YPD to induce endonuclease expression[33] and transformed with a codon-optimized twin-StrepII-tag (5′ TCTGCTTCTGCTTGGTCACATCCACAATTTGAAAAAGGTGG TGG TTCTGGTGGCGGTTCAGGTGGTTCATCTGCTTGGAGTCATCCTCAA TTCGAAAAG 3′)[32] with respective gene-specific overhangs (primers and gene blocks are listed within the Source Data file). Once transformed, clones were screened on YP-galactose plates. The successful gene-target integration of twin-StrepII-tag was validated by PCR.

For Sec13, the integration of the MX4blaster cassette remained unsuccessful unless an additional copy of Sec13 was expressed using a pRS423 overexpression plasmid. To obtain MX4 positive clones, the aforementioned strategy was used. To select the clones, yeast was plated onto His-drop out plates (ForMedium) containing 1 g/L mono-sodium glutamate, 1.9 g/L YNB without amino acids and ammonium sulfate (ForMedium), 3 g/L potassium dihydrophosphate (monobasic), and 300 µg/mL hygromycin B. To insert the StrepII-tag, yeast strains were propagated in the aforementioned media to trigger the loss of the overexpression plasmid. The insertion of the C-terminal StrepII-tag was validated by PCR. Additionally, StrepII-positive strains were plated on His-Drop out plates to ensure the *HIS*-marker removal and that the strains became inviable on His-Drop out plates again.

CCS1-mutants were obtained by transforming the BY4741(*nsp1-strepII*, *nup57::MX4*) with a pRS316 (containing a *tef1*-promoter and a *cyc1*-terminator) encoding an additional copy of *nup57*. Transformants were plated on synthetic complete plates without uracil (SC -Ura). Subsequently, positive clones were propagated for the homologous recombination of the endogenous *nup57* with

alternated CCS1 according to the description above. Clones were selected on SC -Ura supplemented with galactose. pRS316 was removed by plating the strains on 1 g/L 5′FOA (US biological life sciences) plates.

For growth phenotyping, yeast strains were incubated overnight in YPD. On the next day, OD(600) was determined and set to OD(600) of 1 using YPD. Serial dilution in a ratio of 1:10 was prepared in YPD and spotted onto YPD plates. Strains were grown at indicated temperatures. Similarly, CCS1-mutants were plated on SC plates without uracil (-Ura) or supplemented with 1 g/L 5′FOA plates analogous to the spotting assay as described above. SC -Ura plates were incubated for 2 days, while SC 5′FOA plates were incubated for 5 days at 30 °C.

**RIP-qPCR experiments**. The protocol for the RIP-qPCR experiments is an adaptation of the previously published methods[4,6]. For RIP-qPCRs, overnight cultures were grown in YPD. These cultures were used to set 400 mL of YPD to an OD(600) of 0.035. The expression cultures were grown at 30 °C, 160 rpm, and cultured to an OD(600) of 0.5–0.6. Then, cultures were harvested by rapid filtration onto nitrocellulose membrane (0.45 µm; Bio-Rad) and cells were flash-frozen in liquid nitrogen.

Once frozen, cells were supplemented with 1.4 mL of frozen high salt lysis buffer (20 mM Hepes-KOH, pH 7.5, 500 mM KCl, 20 mM MgCl$_2$, 1 mM PMSF, 0.01 % IGEPAL, and cOMPLETE EDTA-free protease inhibitor (Roche), 0.1 mg/ mL CHX (Sigma-Aldrich) or 0.01 mg/mL puromycin (Sigma-Aldrich)). Cells were disrupted under cryogenic conditions using the CryoMill (Retsch) at 30 Hz for 2 min.

The lysate was thawed and transferred into 1.5 mL tubes. The crude lysate was cleared at 15,000 $g$ at 4 °C for 3 min. Afterward, the cleared supernatant was loaded onto equilibrated Streptactin resin (IBA) supplemented with 60 µL of BioLock (IBA) to prevent unspecific binding and 0.1 U/µL Ribolock (Invitrogen) to inhibit RNA decay. The lysate was incubated on the beads by end-to-end mixing at 4 °C for 1 hr. Then, beads were subjected to subsequent wash steps. Briefly, beads were centrifuged at 500 $g$ and 4 °C for 5 min. Supernatant was removed and beads were washed 3-times with 1 mL of wash buffer A (20 mM Hepes-KOH, pH 7.5, 140 mM KCl, 20 mM MgCl$_2$, 0.01 % IGEPAL and cOMPLETE EDTA-free protease inhibitor, 0.1 mg/mL CHX or 0.01 mg/mL puromycin) for 1 min by end-to-end mixing, followed by two washes (1 min, 4 min) with wash buffer B (20 mM Hepes-KOH, pH 7.5, 140 mM KCl, 20 mM MgCl$_2$, 0.05 % IGEPAL and cOMPLETE EDTA-free protease inhibitor, 0.1 mg/mL CHX or 0.01 mg/mL puromycin). After the washes, the beads were resuspended in 500 µL of 10 mM Tris-HCl, pH 8.0.

RNA was extracted by adding 40 µL of 20 % SDS and the addition of 750 µL of pre-warmed phenol-chloroform-isoamyl alcohol (PCI, 65 °C, Invitrogen). This mixture was then incubated at 65 °C, 1,400 rpm for 5 min followed by snap cooling on ice for 10 min. Next, extractions were centrifuged at 15,000 $g$ for 10 min and the aqueous phase was again subjected with 750 µL of PCI. This time, the extraction was performed at room temperature and occasional vortexing for 5 min. The centrifugation was repeated. Finally, residual PCI was removed by a diethyl ether wash, and the remaining organic solvent was evaporated in a Speedvac (Eppendorf).

RNA was precipitated by adding 3 M NaOAc, pH 5.5 to reach a final concentration of 0.3 M, 2.5 µL Glycoblue (Invitrogen), and equivalent amounts of isopropanol. Precipitates were placed into the −80 °C freezer overnight. Samples were centrifuged at 15,000 $g$ and 4 °C for 90 min. The resulting pellet was washed in 70 % EtOH, dried in a Speedvac (Eppendorf), and resuspended in 20 µL of 10 mM Tris-HCl, pH 8.0. Typically, precipitations yielded 150–250 ng/µL of RNA.

For reverse transcription, 500 ng of RNA were applied and cDNA was synthesized according to the manufacturer's instructions of the VILO Reverse Transcription kit (Invitrogen) including the optional ezDNase step.

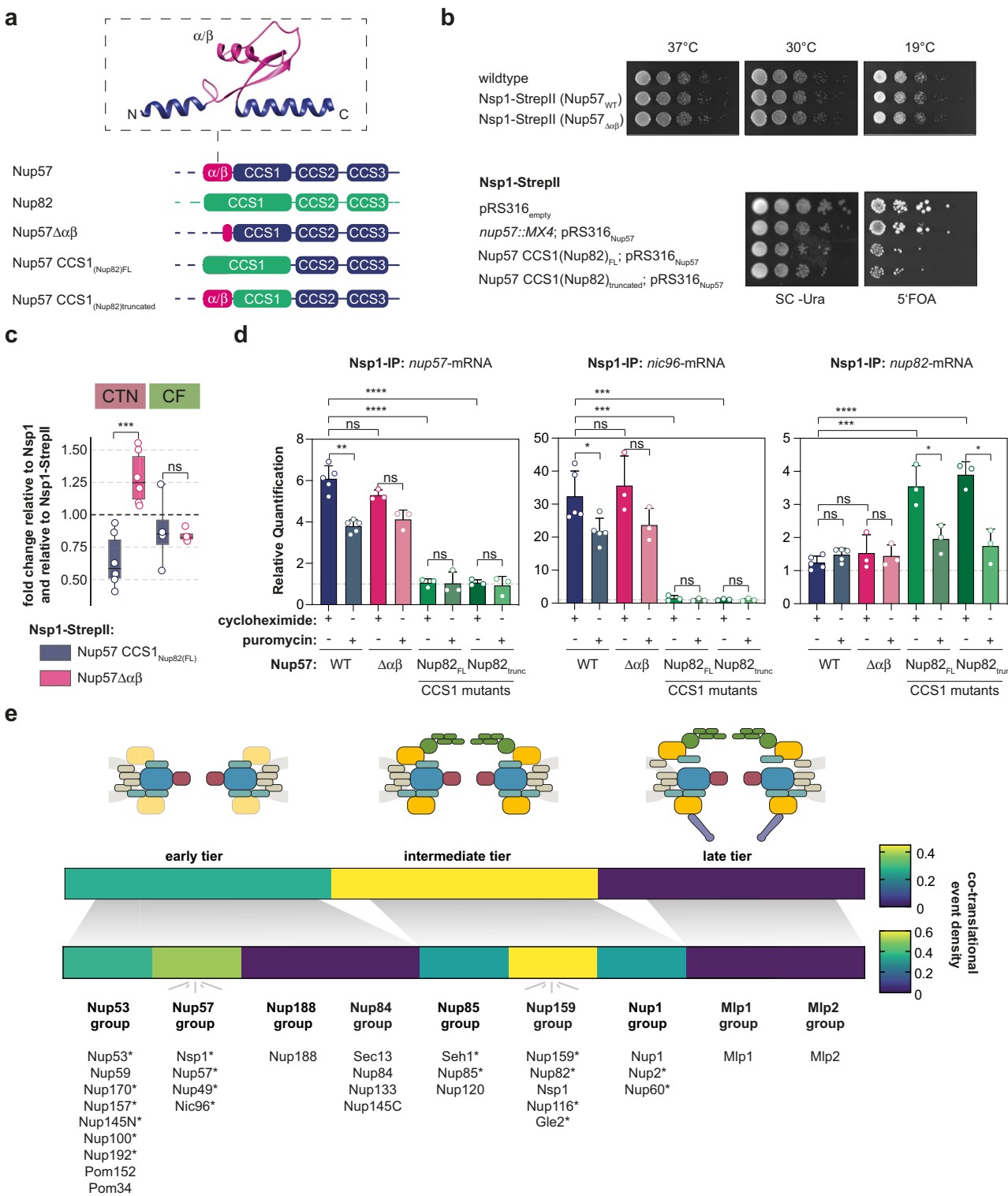

Real-time qPCR was conducted using the TaqMan Fast and Advanced Master Mix (Applied Biosystems) according to the manufacturer's manuscript. FAM-labelled qPCR probes were purchased from Applied Biosystems as specified in Supplementary Table 1. The qPCR was performed using the QuantStudio 5 cycler (Applied Biosystems, 50 °C: 2 min, 95 °C: 2 min; 40 cycles: 95 °C: 0:01 min, 60 °C: 0:20 min). Images were taken every cycle within the annealing/extension step. All qPCR assays were performed in technical triplicates and each experiment was analyzed using the QuantStudio analysis software (v1.5.1). Briefly, quality assessment was performed within the QuantStudio software and if suggested by the software individual technical replicates were omitted. Experiments in which two points of a technical were omitted did not pass our quality control filter. These few cases are also indicated in the Source Data file.

**Selective ribosome profiling.** The selective ribosome profiling experiments were conducted according to previously published protocols by the Bukau lab[4,58]. Each biologically independent replicate was obtained from 800 mL of yeast cultures grown in YPD to an OD(600) of 0.5–0.6 in analogy to the RIP-qPCR experiment. Harvest was performed as previously described for the RIP-qPCR experiments. Subsequently, cells were lysed in 3 mL of ribosome profiling buffer (20 mM Hepes-KOH pH 7.5, 140 mM KCl, 10 mM MgCl$_2$, 1 mM PMSF, 0.01 % IGEPAL, 0.1 mg/mL CHX, 1 tablet of cOMPLETE protease inhibitor per 50 mL) using the cryo-mill (30 Hz, 2 min).

Then, the lysate was thawed and centrifuged at 15,000 g at 4 °C for 3 min. The absorbance at 260 nm of a 1:100 dilution was measured to determine the amount of RNase I (Ambion). Here, 20 U of RNase I per A260 was applied to convert

**Fig. 6 Mutational dissection of the co-translational Nsp1-Nup57 interaction by deletion and domain chimeric mutants. a** Structural scheme of the coiled-coil segments (CCS) of Nup82, Nup57, and Nup57-mutants. The structure depicts CtNup57 from PDB: 5CWS[16]. **b** Growth phenotyping of Nup57 (wildtype) and Nup57-mutants. Deletion of the alpha-beta domain does not impair growth under permissive temperature (*top*). However, the substitution of CSS1 leads to a mild growth defect after removal of a *nup57* rescue copy (*bottom*). **c** Visualization of quantitative mass spectrometry analysis of Nsp1-affinity purifications from wildtype or Nup57-mutant strains. Fold-changes of CTN: central transport Nups (Nup57, Nup49, Nic96) and CF: cytoplasmic filaments (Nup159, Nup82). Fold changes of the respective proteins are indicated as dots. The black dashed line represents abundance of CTN/CF components in a Nsp1-StrepII pull down in wild-type background. Collapsed box plots show median of CTN or CF components with top and bottom reflecting the interquartile range. Whiskers represent 1.5 times the interquartile range. Analysis generated from two biologically independent pull downs ($n = 2$). ***$p = 0.0003$ for the comparison of the CTN-component between the Nsp1-StrepII pulldown in the Nup57Δαβ- and the Nup57 CSS1$_{Nup82(FL)}$-background. **d** RIP-qPCR analysis of Nsp1-IP experiments in wildtype and Nup57-mutants. The Nup57 alpha-beta deletion mutant did not abolish co-translational interactions of Nsp1. In contrast, chimeric mutants abolish the signal for *nup57*-mRNA and *nic96*-mRNA and result into a promiscuous enrichment signal for *nup82*-mRNA. Bar plots of RIP-qPCR data depict mean ± SD. Nsp1-StepII RIP-qPCR experiments in wildtype background were performed with $n = 5$ and for the three respective mutant backgrounds with $n = 3$. **$p = 0.0019$ for Nsp1-StepII (Nup57-WT; *nup57*-mRNA); *$p = 0.0109$ for Nsp1-StepII (Nup57-WT; *nic96*-mRNA); *$p = 0.0352$ for Nsp1-StepII (Nup57 CSS1$_{Nup82(FL)}$; *nup82*-mRNA); *$p = 0.0346$ for Nsp1-StepII (Nup57 CSS1$_{Nup82(trunc)}$; *nup82*-mRNA); ***$p = 0.0005$ for the comparison of Nsp1-StepII (Nup57-WT, *nic96*-mRNA, cycloheximide) and Nsp1-StepII (Nup57 CSS1$_{Nup82(FL)}$/ CSS1$_{Nup82(trunc)}$, *nic96*-mRNA, cycloheximide); ***$p = 0.0002$ for the comparison of Nsp1-StepII (Nup57-WT, *nup82*-mRNA, cycloheximide) and Nsp1-StepII (Nup57 CSS1$_{Nup82(FL)}$, *nup82*-mRNA, cycloheximide); ***$p < 0.0001$ for the comparison of Nsp1-StepII (Nup57-WT, *nup57*-mRNA, cycloheximide) and Nsp1-StepII (Nup57 CSS1$_{Nup82(FL)}$/ CSS1$_{Nup82(trunc)}$, *nup57*-mRNA, cycloheximide) and ***$p < 0.0001$ for the comparison of Nsp1-StepII (Nup57-WT, *nup82*-mRNA, cycloheximide) and Nsp1-StepII (Nup57 CSS1$_{Nup82(trunc)}$, *nup82*-mRNA, cycloheximide). **e** Scheme visualizing co-translational assembly events with respect to the order of the interphase assembly pathway as proposed by Onischenko et al.[29] The heat map includes co-translational events discovered in this study and from Lautier et al.[31] Asterisk (*) marks co-translationally assembling Nups. ns $p > 0.05$, *$p < 0.05$, **$p < 0.01$, ***$p < 0.005$, ****$p < 0.001$ (Two-sided, paired $t$-test for samples for the treatments within one genotype and unpaired, two-sided t-test between the different genotypes). Source data for RIP-qPCR in panel **b**–**d**, are provided as a Source Data file. IP immunoprecipitation, CTN central transport Nups, CF cytoplasmic filaments, CCS coil-coiled segment.

polysomes into monosomes. RNase I was incubated by end-to-end mixing for 20 min at 4 °C. The reaction was quenched by adding 200 U Superase·In (Invitrogen). Then, ribosomes were pelleted using a 25% sucrose cushion (20 mM Hepes-KOH pH 7.5, 140 mM KCl, 10 mM MgCl$_2$, 25% w/v sucrose, 0.01% IGEPAL, 0.1 mg/mL CHX, 1 tablet of cOMPLETE protease inhibitor per 50 mL) at 150,000 *g* for 2.5 h. The ribosomal pellets were resuspended in 1 mL of wash A containing 10 U DNase I (RNase-free, Thermo Scientific) and 30 μL BioLock (IBA). 100 μg of RNA was taken for the total translatome library. The remainders were applied to 250 μL of pre-equilibrated Streptactin sepharose (IBA). The pull-down and RNA precipitation was performed as stated in the RIP-qPCR experiments.

Precipitated RNA was resuspended in 20 μL of TE buffer and supplemented with equal amounts of 2 x RNA loading dye (Thermo Scientific). In parallel, the RNA marker was prepared by mixing the low range RiboRuler (Thermo Scientific) with 200 nM synthetic 5′FAM labeled 34-mer, 30-mer, 28-mer, and 26-mer. Sequences of the custom-made RNAs are listed in Supplementary Table 2.

Both were denatured at 80 °C for 2 min and put back on ice. 15 % denaturing PAGE (Carl Roth) was prepared, prewarmed for 1 h at 16 W and then loaded. The gels were run at 16 W for 3.5–4 h until the bromophenol blue emerged. Afterward, the gels were stained using SybrGold (Invitrogen) and imaged using the Amersham Typhoon (GE Healthcare). The area between 26 and 34 nt were excised and crushed. The RNA was eluted in 500 μL of Tris-HCl pH 8.0 at 70 °C for 10 min while shaking at 1,400 rpm. Elutant was separated from gel pieces by putting them through a Spin-X cellulose acetate column with a pore size of 0.22 μm (Corning). The RNA was precipitated by supplying 50 μL 3 M NaOAc pH 5.5, 2.5 μL Glycoblue co-precipitation agent (Invitrogen), and 500 μL isopropanol.

The purified RNA was initially dephosphorylated in 1 x FastAP buffer containing 2 U FastAP (Thermo Scientific) and 20 U RiboLock (Invitrogen). The reaction was incubated for 15 min at 37 °C and 600 rpm and immediately heat-inactivated at 75 °C for 5 min. The 5′ ends were phosphorylated by 20 U polynucleotide kinase (PNK; NEB) by adding 1 mM ATP (Thermo Scientific), 1 x PNK buffer (NEB), and 20 U RiboLock. The reaction was incubated for 30 min at 37 °C.

Afterward, RNA integrity and concentration were checked using the RNA Pico 6000 Assay Kit of the Bioanalyzer 2100 system (Agilent Technologies). Small RNA libraries were prepared from 1 ng of RNA using the NEXTflex Small RNA-seq Kit v3 (Perkin Elmer). The size distribution of the libraries was assessed on a Bioanalyzer with a DNA High Sensitivity kit (Agilent Technologies), and concentration was measured with the Qubit DNA High Sensitivity kit in the Qubit 2.0 Fluorometer (Life Technologies). Subsequently, libraries that passed the QC step were pooled in equimolar amounts and the final pool was purified with SPRI select beads with a 1.3x ratio (Beckman Coulter). The final pool was loaded on the Illumina sequencer NextSeq500 High output and sequenced uni-directionally, generating ~500 million reads, each 85 bases long.

**Polysome profiling.** Five and 45% sucrose (w/v) was dissolved in 20 mM Hepes-KOH pH 7.5, 140 mM KCl, 10 mM MgCl$_2$, 0.01 % IGEPAL, 0.1 mg/mL CHX and 1 tablet of cOMPLETE protease inhibitor per 50 mL. Gradients were mixed in thin-wall polypropylene tubes (Beckman, 331372) using a gradient mixer (BioComp) and equilibrated overnight at 4 °C. RNA concentration of the cleared lysate was measured by nanodrop and 500 μg of this RNA was loaded onto the gradient and run for 2.5 hr at 220,000 *g* and 4 °C in an SW41-rotor (Beckman). Gradients were then run at 850 μL/min in a density gradient fractionation system (Teledyne Isco), chased by 60 % sucrose in water. RNA absorbance at 254 nm was continuously measured using a UA-6 detector with the sensitivity setting 2.

**Analysis of Nsp1-subcomplexes.** One liter of yeast culture was set to an OD(600) 0.05 and grown to OD(600) 1.2–1.4 at 30 °C and 130 rpm in baffle flasks. Cells were harvested by centrifugation, washed in ice-cold PBS and resuspended in Hepes-NB (20 mM Hepes-KOH pH 7.5, 150 mM NaCl, 50 mM K(OAc), 2 mM Mg(OAc)$_2$, 1 mM DTT, 5% glycerol, 0.01% (v/v) IGEPAL, 1 mM benzamidine, 1 tablet/50 mL of cOmplete EDTA-free protease inhibitor, and 1 mL/50 mL BioLock (IBA)) in adaptation to Fischer et al. Resuspension was frozen drop-wise in liquid nitrogen and lysed using the cryo-mill (30 Hz, 2 min).

The cell lysate was thawed and cell debris was pelleted by ultracentrifugation (35,000 *g*, 20 min, 4 °C). The supernatant was applied to 500 μL bed volume of Streptactin sepharose resin and incubated for 1 h on a rolling mixer at 4 °C. The resin was washed with 4 × 5 mL of Hepes-NB. Protein was eluted in Hepes-NB supplemented with 20 mM D-desthiobiotin (IBA) in four elution steps (3 × 350 μL and 1 × 500 μL) each time incubating the resin 5 min with the elution buffer.

To avoid unnecessary dilution of the elution fractions, the first fraction was omitted. After elution, protein concentration was determined by measuring the absorbance at 280 nm. 500 μL of elution was immediately supplemented with 20 mM TCEP and incubated for 30 min at 37 °C. Next, the pull-downs were subsequently alkylated using 20 mM iodoacetamide (IAA) incubated in the dark for 20 min at room temperature and further processed by adding 12 % aqueous phosphoric acid to obtain a final concentration of 1.2 % of phosphoric acid.

Pre-processed pull-downs were mixed with S-trap binding buffer, transferred to S-trap ProtiFi plates (ProtiFi), and treated according to the manufacture's protocol. Finally, the protein was converted into peptides using a 1:100 Trypsin: protein ratio by supplementing the corresponding amount of Trypsin in 125 μL of digestion buffer that was added to each condition. Trypsin digest was carried out overnight at 4 °C.

Before elution, 80 μL of digestion buffer was added to each well of the S-trap digestion plate and eluted in an OASIS elution plate (Waters). Next, 80 μL of 0.2% of aqueous formic acid was added per well and elution was repeated. Finally, 80 μL of aqueous acetonitrile (ACN) containing 0.2% formic acid was applied and peptides were recovered. The eluted peptides were transferred and solvents were evaporated in a speed vac (Eppendorf). Dried peptides were resolved in 80 μL of HPLC water. 20 μL of these peptides were then subjected for peptide concentration assays (Thermo Scientifc). The remaining peptides were cleaned up using the OASIS desalting plates.

Dried peptides were reconstituted in 5% acetonitrile (ACN) with 0.1% formic acid (FA). Peptides were loaded onto a C$_{18}$-CoAnn trapping column (particle size 3 μm, L = 20 mm) and separated on a C$_{18}$-CoAnn analytical column (particle size

= 2 μm, ID = 75 μm, L = 50 cm, CoAnn Technologies, LLC, Richland, USA) using a nano-HPLC (Dionex U3000 RSLCnano) at a temperature of 55 °C.

Trapping was carried out for 6 min with a flow rate of 6 μL/min using a loading buffer (100% $H_2O$ with 0.05% trifluoroacetic acid). Peptides were separated by a gradient of water (buffer A: 100% $H_2O$ and 0.1% FA) and acetonitrile (buffer B: 80% ACN, 20% $H_2O$, and 0.1% FA) with a constant flow rate of 250 nL/min. The gradient went from 4% to 48% buffer B in 90 min. All solvents were LC-MS grade and purchased from Riedel-de Häen/Honeywell (Seelze, Germany).

Eluting peptides were analyzed in data-dependent acquisition mode on a Fusion Lumos mass spectrometer (ThermoFisher Scientific) coupled to the nano-HPLC by a Nano Flex ESI source. MS1 survey scans were acquired over a scan range of 350–1400 mass-to-charge ratio (m/z) in the Orbitrap detector (resolution = 120k, automatic gain control (AGC) = 2e5, and maximum injection time: 50 ms). Sequence information was acquired by a "ddMS$^2$ OT HCD" MS2 method with a fixed cycle time of 2 s for MS/MS scans. MS2 scans were generated from the most abundant precursors with a minimum intensity of 3e4 and charge states from two to five. Selected precursors were isolated in the quadrupole using a 1.4 Da window and fragmented using higher-energy C-trap dissociation (HCD) at 30 % normalized collision energy. For Orbitrap MS2, an AGC of 1e4 and a maximum injection time of 54 ms were used (resolution = 30k). Dynamic exclusion was set to 30 s with a mass tolerance of 10 parts per million (ppm). Each sample was measured in duplicate LC-MS/MS runs.

MS raw data were processed using the MaxQuant software (v1.6.6.0) with customized parameters for the Andromeda search engine. Spectra matched to a *Saccharomyces cerevisiae* database downloaded from UniProtKB (April 2021), a contaminant and decoy database, with a minimum Tryptic peptide length of seven amino acids and a maximum of two missed cleavage sites. Precursor mass tolerance was set to 4.5 ppm and fragment ion tolerance to 20 ppm, with a static modification (carboxyamidomethylation) for cysteine residues. Acetylation on the protein N-terminus and oxidation of methionine residues were included as variable modifications. A false discovery rate (FDR) below 1% was applied at protein, peptide, and modification levels. The "match between runs" option was enabled and only proteins identified by at least one unique peptide were considered for further analysis.

All proteomics data (including acquisition and data analysis parameters) associated with this manuscript have been deposited at the ProteomeXchange Consortium (http://proteomecentral.proteomexchange.org) via the PRIDE partner repository[59] under accession codes PXD030626 and PXD028413.

For comparing the co-enrichment of CTN and CF components, iBAQ values were extracted from ProteinGroups- output from MaxQuant Analysis. iBAQ profiles were median-normalized and then normalized to the iBAQ profile of Nsp1. Finally, iBAQ values from Nsp1-StrepII pull-downs in Nup57 CCS1$_{Nup82(FL)}$ and Nup57 without alpha-beta domain were divided by respective iBAQ values in Nsp1-StrepII condition in wildtype background, stratified by the respective batch. Values for the tagged proteins, underlying Fig. 6c, were compared using a two-sided unpaired *t*-test, when deemed possible (n > 4).

**Protein analysis**. Four microliter of the crude lysate or RNCs (ribosome pellet) (representing 0.4%) in 1 x NuPAGE loading dye (Invitrogen) and 10 μL of boiled beads (representing 4%) in 1 x NuPAGE loading dye were loaded onto NuPAGE Bis-Tris gels (MW < 100 kDa) (Invitrogen) or Tris-Glycine gels (Bio-Rad; MW > 100 kDa) and run at 160 V for 50 min. Protein was transferred onto 0.45 μm TransBlot Turbo nitrocellulose (Bio-Rad) using the High MW setting of the TurboBlot system (Bio-Rad) according to the manufacturer's procedure. Membranes were blocked in 5% milk in TBS-T (0.02% Tween-20) for 1 hr at room temperature under gentle shaking. Primary antibody (1:5000) was added and incubated overnight at 4 °C while constant shaking. Next, membranes were washed in TBS-T and a secondary antibody (1:10,000) was applied. The membrane was stained for 1 h at room temperature. Before visualization by ECL developing solution (Bio-Rad), membranes were washed again. Membranes were imaged using the Chemidoc (Bio-Rad). Antibodies used in this study are listed in Supplementary Table 3.

Fas1- and Fas2-IPs were stained using Instant Blue (abcam) according to the manufacturer's protocol. For Nsp1 pull-downs, NuPAGE Bis-Tris gels were stained using the SilverQuest Silver Staining Kit (Invitrogen).

**Data processing**. Sequencing reads were processed according to the guidelines published in Galmozzi et al.[58]. In brief, reads were cleaned and trimmed using cutadapt (v2.3)[60]. Sequences mapping to *Saccharomyces cerevisiae* noncoding RNA (R64-1-1.ncrna) were discarded and the remaining reads were mapped to *Saccharomyces cerevisiae* (R64-1-1) genome using tophat2 (v2.0.10). To analyze and assign ribosome positions scripts from the script suite (https://doi.org/10.5281/zenodo.2602493) were used.

The number of reads per genomic position was extracted using script A and was used as input for subsequent analysis using in-house MATLAB scripts (v9.7.0.1296695 (R2019b) Update 4). The scripts combined data from different replicates and used gliding averages to evaluate the enrichment within a given sequence window.

Additionally, we used Script C and D[58] to generate footprint distribution plots and the total enrichment (TE) file.

For the SeRP analysis, a Limma-analysis[61] of the TE was conducted considering groupings of IP-experiments vs. total-RNA-experiments. *p*-values from respective fittings were adjusted using the Benjamini-Hochberg method.

Structures were analyzed using UCSF Chimera (v1.15).

**Statistics and Reproducibility**. Data in figures was illustrated as mean with corresponding standard deviation (SD) using GraphPad Prism (v9.0.0). Dashed lines in qPCR graphs represent wildtype background levels (no bait) determined for cycloheximide and puromycin, respectively. Significance levels of qPCRs for one mRNA obtained from the same bait under cycloheximide and puromycin treatments was determined by applying a two-sided Student's *t*-test for paired samples by assuming a normal distribution of the data unless otherwise stated (ns: $p > 0.05$; $*p < 0.05$; $**p < 0.01$; $***p < 0.001$). Comparison of Nsp1-StrepII pull downs in Nup57(wild-type) and Nup57-mutant strain background was conducted using a two-sided unpaired *t*-test (ns: $p > 0.05$; $*p < 0.05$; $**p < 0.01$; $***p < 0.001$, $****p < 0.0001$). Co-translational mRNA enrichment was assigned based on the two following parameters: (I) average signal of cycloheximide treated lysate ≥1.5 and (II) a statistically significant signal decrease upon puromycin treatment. For mass spectrometry, Nup49 and Nup57 and Nup159 and Nup82 were grouped into CTN and cytoplasmic filaments, respectively. Significance levels were calculated in respect to Nsp1 by a two-sided, unpaired Student's *t*-test (ns: $p > 0.05$; $*p < 0.05$; $**p < 0.01$; $***p < 0.001$).

Western Blot analysis, Coomassie and Silver Staining of the pull-downs shown in Supplementary Fig. 2a, d, 4c and 8a were performed at least twice. The denaturing PAGE for ribosome profiling (Supplementary Fig. 4d) was run once for the respective replicate. Growth assays as shown in Fig. 6b were repeated twice.

Co-translational event density (Fig. 6e) was calculated by dividing the number of co-translational events by the number of proteins per group. Secondary binding and collective binding as previously observed for the translation-dependent interactions of Gle2 - nup82 and CTN - nic96 was considered as one event.

**Strains**. Yeast strains used in this study are listed in Supplementary Table 4.

**Reporting summary**. Further information on research design is available in the Nature Research Reporting Summary linked to this article.

## Data availability

The underlying RIP-qPCR data, uncropped images (e.g. Western Blots, growth assays), polysome profiles, Limma-results, underlying mass spectrometry data and primers which were generated in this study are provided in the **Source Data file**. The selective ribosome profiling data used in this study are available in the European Nucleotide Archive database under accession code PRJEB46361 and PRJEB50305. The mass spectrometry data generated in this study have been deposited in the PRIDE database under accession codes PXD030626 and PXD028413.

The previously published structures for 4XMM [https://doi.org/10.2210/pdb4XMM/pdb] (Fig. 3a: Seh1-Nup85 dimer within the Nup84 subcomplex), 4BZK [https://doi.org/10.2210/pdb4BZK/pdb] (Fig. 4c, d, f: COPII coat consisting of Sec13-Sec31), 3MZK [https://doi.org/10.2210/pdb3MZK/pdb] (Fig. 4e, f: Sec13-Sec16 complex) and 5CWS [https://doi.org/10.2210/pdb5CWS/pdb] (Figs. 5a and 6a: Central transport Nup-trimer) are accessible at the Protein Data Bank (PDB). The integrative structure of the cytoplasmic filaments[21] is available in the PDB-Dev under accession code PDBDEV_00000010 (Fig. 5d). The electron density map of the NPC[45] is deposited in the Electron Microscopy Data Bank under accession code EMD-10198 (Fig. 1a). Source data are provided with this paper.

## Code availability

MatLab scripts for analysis and plotting of SeRP data were deposited to Zenodo (https://doi.org/10.5281/zenodo.5887401). The script suite for SeRP[58] can be found on Zenodo (https://doi.org/10.5281/zenodo.2602493) and includes the required reference genome files for the coding and non-coding genome of *Saccharomyces cerevisiae* (R64-1-1).

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

## Acknowledgements

We thank Patrick Hoffmann, Jan Provaznik and Florian Wilfling for the fruitful discussion on the manuscript. Additionally, the authors like to thank Georg Stoecklin, Lars Steinmetz, and Britta Brügger for the critical assessment of the project. E.M.S. is funded by the Max Planck Society and an Advanced Investigator award from the European Research Council (grant 743216). J.D.L. and G.H. acknowledge their funding by the Max Planck Society. M.B. acknowledges funding by the Max Planck Society and the European Research Council (724349-ComplexAssembly).

## Author contributions

M.S. conceived the project, designed experiments, performed experiments, analyzed data, and wrote the manuscript. A.B. performed experiments. F.P. designed experiments. J.J.M.L. and N.T.D.d.A performed experiments, analyzed data, and wrote the manuscript. C.M.F. performed experiments. E.K. performed experiments. N. R. analyzed data and wrote the

manuscript. J.B. designed experiments, analyzed data, wrote the manuscript. J.D.L., E.M.S., and K.R.P. supervised the project. G.H. performed modeling and wrote the manuscript. V.B. designed experiments, supervised the project. M.B. conceived the project, designed experiments, analyzed data, supervised the project, and wrote the manuscript.

## Funding

## Competing interests
The authors declare no competing interests.
