## [Peer Review File · Nature Communications]

REVIEWER COMMENTS

Reviewer #1 (Remarks to the Author):

Using a qPCR-based approach to detect cotranslational interactions between multiple members of the nuclear pore complex, Seidel and colleagues confirm several recently published and 4 previously unknown cotranslational assembly reactions. They additionally use selective ribosome profiling (SeRP) to provide nascent polypeptide length-resolved data for some of these cotranslational events. The study i) determines the order of cotranslational assembly events leading to the formation of the central transport Nup (CTN) heterotrimer, and ii) points at structural motifs that mediate cotranslational assembly of the targeted subunits. Furthermore, by testing the cotranslational interaction of moonlighting proteins with their interaction partners in the different complexes, the authors raise the concept that cotranslationally established interactions may introduce a bias of moonlighting proteins for one assembly pathway.

The study is well executed, expands our understanding of the assembly of the nuclear pore complex and raises an interesting concept regarding the effect of cotranslational assembly of moonlighting proteins. However, major conclusions drawn from data presented require additional experimental support.

Major concerns:

- 1) The authors argue that cotranslational assembly of moonlighting proteins counteracts their promiscuous interaction with subunits of other complexes. This is claimed in the title, abstract and at multiple places throughout the text. However, the data presented is not sufficient to establish that model. Cotranslational assembly of a fully synthesised moonlighting protein with a nascent interaction partner of one complex does not prevent its post-translational assembly with another complex, and may not bias the assembly pathway at all. The authors should address these issues by:
 - a. Clarifying how the assembly of fully synthesised Nup53 with nascent Nup170 introduces a bias against its assembly with fully synthesised Nup157 (Line 194, Fig. 2f). In their response the authors should consider that nascent Nup53 itself is engaged by Nup157 and Nup170 with an effect size comparable to the Nup157 RIP-qPCR detecting nup145 mRNA, which they assume is biologically relevant (Fig 2j).
 - b. Better explaining the conclusions drawn on Nup116 assembly with nascent Nup82 (line 213).
 - c. Specifying the meaning of “unwanted complexes” (line 197), considering that both Nup53-Nup157 and Nup53-Nup170 are biologically relevant complexes.

2) The interpretation of the data has to be more consistent. For example, data in figure 4e is presented as cotranslational assembly (line 295), and in figure 2j it is being addressed as weak cotranslational interactions (line 207), both with similar effect size. A similar effect size in results on line 197 and figure 2f is not considered as cotranslational assembly. Furthermore, it is not clear how biologically relevant such small effect sizes really are (e.g. figure 3B Seh1-seh1 interaction).

3) Compared to SeRP, the RIP-qPCR approach seems to generally suffer from high variability and reduced sensitivity, as indicated by the reduced effect sizes in Fig 5d compared to Fig. 4b, Fig. 2d and Fig. 4. This raises the concern that RIP-qPCR data may mask differences (for example between Nup57-WT and Nup57- $\Delta\alpha\beta$) or even generate false negative results (for example in figure 5d). Therefore, the authors should support main claims by selective ribosome profiling, especially for positive results with very small effect sizes.

4) The results section of the text is challenging to follow. The presentation of the results is written in a branched, non-linear development of the scientific discoveries. An example for that is that in figure 2d and e the authors present results regarding the CTN subcomplex. The authors then move to a different subcomplexes in figure 2, different aspect of their work in figure 3, and then back to the CTN subcomplex in figure 4. In which, they describe the pre-existing steps for the results described in figure 2d and e. Readers which are not fluent in the structure of the nucleopore complex might find it confusing. I therefore suggest rethinking the order by which the results are presented.

Minor concerns:

5) Line 143: "In order to elucidate the implementation of co-translation for faithful assembly of the NPC ..." please clarify the meaning of co-translation in this context. (e.g. "co-translational assembly"?)

6) Line 163: The RIP-qPCR data presented does not allow the strong conclusion at this point of the text.

7) Line 184: Nup49 should have been tested as well.

8) Line 233/Fig. S5c-f: Rather than showing individual, pre-selected SeRP profiles, IP/total RPKM enrichments for each mRNA should be presented in a scatter plot to get an overview of the nascent interaction landscape of the protein of interest.

9) Line 259: The authors should support their interpretation that Sec13 may bind sec31 mRNA rather than the Sec31 nascent chain (NC), e.g. by SeRP. Sec31 is a rather large protein with a correspondingly long mRNA and presumably covered with many ribosomes. Since puromycin does not release NCs with 100% efficiency, and retaining a single sufficiently long nascent chain can result in the pull down of a given mRNA via NC-NC interactions, the enrichment using the RIP-qPCR approach may be decreased for long/highly translated mRNAs.

10) Fig. 4g: To validate the cotranslational assembly pathway depicted (Gle2 → Nup116 → Nup82), Gle2 RIP-qPCR data testing for the enrichment of nup82 mRNA should be added.

11) Line 284/Fig. S6: Although generally conceivable, the interpretation that locally decreased translation efficiencies correlate with the assembly onsets as shown in Fig. S6 is questionable. Instead, the authors should test for an increased density of ribosomes before the onset of cotranslational assembly in the ribosome profiling data, indicating decreased local translation speed.

12) Line 346: The authors probably meant to write that “Sec13 might rather associate with Sec31 post-translationally (after, not prior to) the co-translational entanglement of Sec31 with itself”

13) Line 375: The authors should provide data or clarify which of the data shown, and in which manner, support their conclusions about the stoichiometry during CTN formation and its assembly with Nic96.

14) Fig. 5d: Indicate the significance of the differences between samples treated with CHX or puromycin to support the finding that alpha-beta deletion does not affect the cotranslational assembly of Nsp1 on nascent Nic96.

Reviewer #2 (Remarks to the Author):

Seidel et al. employ an affinity purification coupled qPCR method to identify proteins that interact with ribosome bound nascent chain proteins. They elucidate important insights into the biogenesis of multiple subcomplexes of the NPC shedding light on the mystery of how cells avoid forming nonproductive deadend complexes when proteins moonlight in multiple complexes. Through selective ribosome profiling they further analyze at which point in the growing nascent chain complexes form. In the case of Nsp1-Nup57 binding the nascent chain on Nic96 they demonstrate association begins immediately after the binding motif is exposed from the exit tunnel. An important insight comes when examining Seh1-Nup85 nascent chain which surprisingly demonstrates it is not necessarily immediately at the binding site but as in the case of the association begins much farther C-terminal of the interaction site. The results are robust and show order of assemble such as Nsp1 binding the nascent chain of Nup57 prior to the Nsp1-Nup57 subcomplex binding the growing nascent chain of Nic96 as well as discrimination such as Nup53 binding the growing nascent chain of Nup170 but not Nup157. They expand the analysis to show the Gle2-Nup116-Nup159-Nup82 complex binds the growing chain of Nsp1 and thus avoids forming a complex with Nup57-bound Nsp1. These important insights in NPC biogenesis and faithful complex formation were carefully and robustly performed and the paper is beautifully illustrated and the manuscript well written. The senior author should take another look at the literature cited. The paper will be of considerable interest to a broader audience, and I enthusiastically support publication.

The following minor comments are suggestions that the authors may wish to consider including in the revision of their manuscript.

Minor points

1. The authors may want to explicitly explain why they compare cycloheximide stalled ribosomes with puromycin stalled ribosomes which they specify dissociated the nascent chain on line 157.
2. The authors suggest in line 306 “the sequences of the C-terminal coiled-coil segments of Nup57 and Nup82 align well with one another (Figure 5a)” yet reference a domain architecture. It may be more accurate to state the secondary structure or domain architecture aligns well.
3. In the description of the Sec13 and Seh1 interactions with Nup145C and Nup85, respectively, the authors correctly cite Debler et al., Mol Cell for the Seh1-Nup85 interaction, but should also include the citation that originally reported the “domain invasion motif” in the Sec13-Nup145C complex 2 years earlier (Hsia KC, Stavropoulos P, Blobel G, Hoelz A. (2007). Architecture of a coat for the nuclear pore membrane. Cell 131, 1313-26). The citation should be inserted at the end of the following paragraph:

Inspired by the above findings, we wondered whether co-translational events may also specify assembly pathways for moonlighters that are members of multiple protein complexes. Two members of the Nup84-subcomplex, Sec13 and Seh1, are incomplete beta-propellers lacking one blade, that interact with the WD40 domain invasion motifs of Nup85 and Nup145C, respectively (Hsia et al. 2007; Debler et al., 2008).

4. In the description of the Nup116 GLEBS interaction with Gle2 (page 7), the authors may want to consider citing the paper that reports the structure of the complex (Ren Y, Seo HS, Blobel G, Hoelz A. (2010). Structural and functional analysis of the interaction between the nucleoporin Nup98 and the mRNA export factor Rae1. PNAS 107,10406-11). The reference should be included after the following sentence:

The N-terminal GLEBS domain of Nup116 binds to Gle2 and is absent in Nup100 or Nup145N (Bailer et al., 1998; Ren et al., 2010) (Figure 2g).

Reviewer #3 (Remarks to the Author):

The principles that dictate cotranslational assembly of protein complexes—particularly what factors determine which complexes assemble cotranslationally while others don't—remain mysterious. The manuscript by Seidel et al. addresses this broad question by analyzing specific nucleopore subcomplexes using immunoprecipitations of candidate factors coupled with reverse transcription or ribosome profiling. Although the study does not identify hard and fast rules of cotranslational assembly, the thorough and targeted descriptions of different complex behavior will undoubtedly contribute to our understanding of nucleopore assembly and help the field establish trends of cotranslational assembly. The experiments are expertly designed, conducted, and analyzed, and the figures are meticulously assembled. I generally support publication of this study and have only two minor comments listed below.

1. Personally, I did not find Figure 1 or the first two paragraphs of the Results section (lines 105-136 particularly informative or helpful. These points seem almost trivial and could easily be summarized in the introduction, and Figure 1 does not present especially novel theoretical modeling. I am also not sure if I agree with the implication in Figure 1 that each step of an assembly pathway has the same probability of succeeding (or failing). Given that the introduction and discussion section already seem long, this portion of the texts seemed to only delay progression to the true results of the study. I recommend shortening or removing this section.

2. Some parts of the text border on overinterpretation without parallel supporting evidence. In general, the data speak for themselves, and these statements are not necessary. Specific instances listed below:

Line 261-262: the implication of protein binding to the 3'UTR seems premature without further analysis, and because this is the only example of puromycin-independent interaction in the text.

Line 277-279: the statement "...the minimal requirement for co-translational interactions is the initial..." should be reworded or validated by chimeric experiments (e.g. transplanting the CCS1 to a different nascent protein to show sufficiency).

Pages 9-10: the discussion of the two separate Nsp1 complexes heavily rely on previous studies, but the current study doesn't biochemically demonstrate that these distinct subcomplexes can be separated, and the presented data cannot directly support order of assembly. Language around these interpretations, e.g. line 302-303 "...tetramer forms co-translationally first and only subsequently binds to Nsp1" should be carefully curated.

We want to thank all reviewers for the constructive critique and the overall positive assessment of our manuscript! We have revised our manuscript according to their suggestions and included additional mutational analysis and selective ribosome profiling data. Our detailed point by point response follows below.

Reviewer 1:

Major concerns:

1) *The authors argue that cotranslational assembly of moonlighting proteins counteracts their promiscuous interaction with subunits of other complexes. This is claimed in the title, abstract and at multiple places throughout the text. However, the data presented is not sufficient to establish that model. Cotranslational assembly of a fully synthesized moonlighting protein with a nascent interaction partner of one complex does not prevent its post-translational assembly with another complex, and may not bias the assembly pathway at all. The authors should address these issues by:*

We adapted the term 'promiscuous assembly' from previous literature on the assembly of nucleoporin subcomplexes. For example, the characterization of the CTN subcomplex (Nup62 in mammals) revealed promiscuous protein interactions and non-physiological stoichiometry *in vitro* (Sharma et al., 2015; Ulrich et al., 2014) that even had resulted in crystal structures with non-native contacts and oligomerization state (Melčák et al., 2007; Solmaz et al., 2011). The native and physiologically relevant structures (Chug et al., 2015; Stuwe et al., 2015) that turned out compatible with *in vivo* NPC architecture (Kosinski et al., 2016; Lin et al., 2016) comprise only a subfraction of the possible complexes formed *in vitro*. Furthermore, the Nup159 and CTN subcomplexes have homologous scaffolds and one overlapping subunit (Nsp1). Despite a considerable number of combinatorial possibilities, only the Nsp1:Nup159:Nup82 and Nsp1:Nup57:Nup49 heterotrimers are physiologically relevant.

Ulrich and Schwartz (2014) state: *"Furthermore, we observe that eliminating one binding partner can result in the formation of complexes with noncanonical stoichiometry, presumably because unpaired coiled-coil elements tend to find a promiscuous binding partner"* (Ulrich et al., 2014). Their data implies that a non-physiological Nup49:Nup57:Nup49 complex is generated without Nsp1 while removal of Nup49 results in the accumulation of an assembly intermediate (Nsp1-Nup57 heterodimer) :

<https://www.ncbi.nlm.nih.gov/pmc/articles/PMC4004597/figure/F4/>

One of our major objectives was to address how cells cope with the potentially promiscuous assembly of these subunits. We reasoned that co-translational assembly events impose a hierarchical order to the assembly pathway that prevents non-physiological association. Furthermore, co-translational assembly occurs at synthesis intermediates that do not yet expose 'promiscuous' interaction surfaces which are located more towards the C-terminus. The interaction of the 'hidden' C-terminal domain with unwanted interaction partners is thus prevented. The dwell time of such intermediates may even be increased by a reduction in translational speed. In case of Nup57, the assembling coiled-coil domain is C-terminal. It is thus plausible that it cannot form the unwanted Nup49:Nup57:Nup49 complex because it is co-translationally associated with Nsp1 already before it leaves the exit channel of the ribosome. In case of the Nup159 subcomplex (cytoplasmic filaments) that forms a competing assembly pathway, Nsp1 associates post-translationally, underlining that there has to be a soluble pool of this protein, which is in line with our model.

To further support this concept, we included additional mutational analysis. We substituted the CCS1 domain of Nup57 (Bailer et al., 2001) that engages with Nsp1 co-translationally, with the competing CCS1 of Nup82 (Nup57 domain swap mutant). As expected, this results in a depletion of co-translational engagement of Nsp1 with nascent Nup57. Strikingly, a non-canonical, co-translational interaction of Nsp1 with Nup82 becomes apparent that remained

undetected under wildtype conditions as shown by our SeRP and qPCR data (**Figure 4f and 5d**). We also observed that the assembly of the CTN subcomplex is partially compromised in the swap mutant (**Figure S9b and S9c**). We interpret this observation as a weaker affinity of Nup82 to Nsp1 that becomes co-translationally relevant only once excess of Nsp1 is generated because CTN complex assembly is impaired. This data underlines co-translational assembly can suppress unwanted interactions.

We made an effort to explain this better in the main text and discussion of our manuscript. An alternative title could be “Co-translational assembly orchestrates competing assembly pathways”. We would appreciate the reviewer’s feedback on this alternative.

- a. *Clarifying how the assembly of fully synthesized Nup53 with nascent Nup170 introduces a bias against its assembly with fully synthesized Nup157 (Line 194, Fig. 2f). In their response the authors should consider that nascent Nup53 itself is engaged by Nup157 and Nup170 with an effect size comparable to the Nup157 RIP-qPCR detecting nup145 mRNA, which they assume is biologically relevant (Fig 2j).*

We agree with the reviewer that in case of Nup53 it is not immediately intuitive how the co-translational assembly with nascent Nup170 imposes a bias towards Nup157. In contrast to Nsp1 this cannot be explained by a hierarchical organization of the assembly pathway alone. We think it is still interesting to note that co-translational interactions are detected only within the physiologically relevant pathway. Our data suggest that *in vivo* the affinity of Nup53 for nascent Nup170 is higher as compared to nascent Nup157, which also could bias the assembly pathway. It furthermore agrees with cross-linking data by Kim et al, Nature 2018 that indicate interactions of Nup59 and Nup53 with Nup170 but not Nup157 *in vivo* and the most recent integrative analysis of yeast NPCs that has revealed that Nup53/Nup59 heterodimers interact with Nup170 but not Nup157 (Akey et al., 2021).

<https://www.biorxiv.org/content/10.1101/2021.10.29.466335v2.full.pdf>

The productive assembly may be further supported by the compositional differences in the inner and outer core modules namely Nup192/Nup188. We have nevertheless down-toned the respective sentence in the main text as follows: “*Interactions with both, Nup157 and Nup170 were reported in vitro (Onischenko et al., 2009), but integrative modelling highlights the physiological relevance of the interaction with Nup170 (Akey et al., 2021; Kim et al., 2018). We found that Nup53 co-translationally binds to nascent Nup170 but not Nup157 (Figure 1f). In our RIP-qPCR experiment, Nup170 and Nup157 did display significant signal reduction for nup53-mRNA upon puromycin treatment. However, the small effect size (<1.5) and additional SeRP experiments (Figure S6) do not support that this reduction was due to co-translational engagement. In sum, it is interesting to note that co-translational assembly events are detected only on the physiologically relevant pathway.*”

Regarding the effect sizes, we agree that the differences are not extremely strong in this particular case but they are elevated for the relevant interactions. In response to point 2) of the reviewer we have introduced a consistent threshold for all data in the revised version, namely a 1.5-fold enrichment within the CHX condition and a statistically significant decrease in signal upon puromycin treatment (see below for more detail). The relevant effect sizes are discriminative, namely 1.9-fold for Nup53 with nascent Nup170 but only 1.2-fold for Nup53 with nascent Nup157 (also no significant decrease with puromycin).

The inverse interactions are clearly below threshold, namely 1.1-fold for Nup157 with nascent Nup53 and 1.3-fold for Nup170 with nascent Nup53. In case of Nup157 that interacts with nascent Nup145 the enrichment is 1.47-fold and thus just under threshold, and the interactions with the paralogous Nup100 (1.0-fold) and Nup116 (1.0-fold) are clearly below threshold. This is further underscored by additional SeRP data that also did not reveal any signal for those interactions (**Figure S6**, see below). Data interpretation and analysis is thus consistent in the revised version of our manuscript.

- b. *Better explaining the conclusions drawn on Nup116 assembly with nascent Nup82 (line 213).*

Nup82 was shown to bind to the paralogous Nup145N, Nup100 and Nup116 *in vitro* (Fischer et al., 2015; Yoshida et al., 2011), but *in vivo* analysis of *Saccharomyces cerevisiae* NPC suggested that the interaction with Nup116 is physiologically most relevant (Allegretti et al., 2020).

We propose that the Nup116 specificity for the CF (cytoplasmic filament) complex (also called Nup159 complex) arises from the co-translational attachment of nascent Nup116 with Gle2 and the consecutive co-translational binding to Nup82 (summarized in **Fig 4g**). We edited the main text to make this clearer: *“The C-terminal autoproteolytic domains (APD) of Nup145N, Nup100 and Nup116 bind to the cytoplasmic filament protein Nup82 in vitro (Fischer et al., 2015; Yoshida et al., 2011). However, integrative (Kim et al., 2018) and in situ structural analysis (Allegretti et al., 2020) stresses the physiological relevance of the interaction with Nup116. Our RIP-qPCR data reveal that Gle2, Nup116 and Nup82 form a hierarchical co-translational assembly chain (Figure 1k, S3c) that may determine Nup116 specificity for the cytoplasmic filaments.”*

- c. *Specifying the meaning of “unwanted complexes” (line 197), considering that both Nup53-Nup157 and Nup53-Nup170 are biologically relevant complexes.*

Please see response to point a. above. We have edited the text for more clarity.

2) *The interpretation of the data has to be more consistent. For example, data in figure 4e is presented as cotranslational assembly (line 295), and in figure 2j it is being addressed as weak cotranslational interactions (line 207), both with similar effect size. A similar effect size in results on line 197 and figure 2f is not considered as cotranslational assembly. Furthermore, it is not clear how biologically relevant such small effect sizes really are (e.g. figure 3B Iseh1 interaction).*

We have decided to remove statements referring strong or weak co-translational signals as the effect size depends on: i) ORF length (the longer an ORF, the stronger the signal), ii) the position of the onset/ interaction site within the ORF (the more N-terminal, the stronger the signal) and iii) potential implications in pull down efficiency, for example solubility of the bait. In case of Nup53, we reason that small effect size may be due to the rather C-terminal interaction sites and limited solubility due to its amphipathic helix. In contrast, N-terminal signals of e.g. the CTN with nascent Nic96 (**Fig 1e**) and Gle2 with nascent Nup116 (**Fig 1h**) yield better effect sizes likely due to higher ribosome occupancy along the ORF.

As mentioned above, we now have introduced an arbitrary but consistent threshold of 1.5-fold above background (with cycloheximide above a non-tagged strain). In addition, a statistically significant reduction of signal with puromycin has to be observed. These parameters are consistently used throughout the manuscript. They are in line with SeRP analysis and the data by Lautier et al, 2021, where relevant. We have edited the methods part as follows: *“Co-translational mRNA enrichment was assigned based on the two following parameters: (I) average signal of cycloheximide treated lysate ≥ 1.5 and (II) a statistically significant signal decrease upon puromycin treatment.”*

Regarding Seh1-*seh1* interaction that the reviewer refers to, the reduction observed with puromycin borderlines for significance. However, the effect size observed above background is very small and not even close to 1.5-fold. Thus, only one of the two criteria is fulfilled. Consistently, this interaction has not been observed in SeRP (**Fig 2c**) and thus can be neglected.

3) Compared to SeRP, the RIP-qPCR approach seems to generally suffer from high variability and reduced sensitivity, as indicated by the reduced effect sizes in Fig 5d compared to Fig. 4b, Fig. 2d and Fig. 4. This raises the concern that RIP-qPCR data may mask differences (for example between Nup57-WT and Nup57- $\Delta\alpha\beta$) or even generate false negative results (for example in figure 5d). Therefore, the authors should support main claims by selective ribosome profiling, especially for positive results with very small effect sizes.

RIP-qPCR is performed under high salt lysis conditions in order to achieve nascent chain dissociation upon puromycin treatment (Blobel and Sabatini, 1971). It does not include an additional enrichment of polysomes prior to the immunoprecipitation. Therefore, signal could certainly be masked by unspecific binding of untranslated mRNA to the beads and one cannot entirely exclude false-negatives, as for many other techniques.

Nevertheless, RIP-qPCR has advantages. It enables the screening for interactions in a high throughput manner. It is always accompanied with a negative control which allows to distinguish between co-translational interactions and noise. Therefore, the potentially reduced sensitivity can be counteracted by testing more conditions. This is a valuable complementation to SeRP that is experimentally very demanding and costly.

For reasons explained in response to point 2) above, the effect size for the enrichment may be variable across the different baits but also mRNAs. However, the combination with the puromycin treatment provides a very stringent filter for translation dependency.

We have strengthened the analysis shown now in **Figure 5d** by additional replication and mutations. Taken together the results are very clear: With Nsp1 as bait, wildtype and Nup57 alpha-beta deletion mutant enrich for the *nup57* and *nic96* but not the *nup82* mRNAs. This behavior is reversed in both of the swap mutants.

We analyzed the following additional targets by SeRP, as suggested by the reviewer: Sec13, Sec31, Nup170, Nup157 and Nup53 (**Figure S4c and S6**). The Nup53 pulldown did not pass our quality criteria (bait not detected by Western Blotting; **Figure S4c**), which we attribute to reduced solubility due to the amphipathic helix contained in Nup53. The Sec13 and Sec31 analysis revealed very striking insights into the role of the insertion blades contained in Sec31 and Sec16 (**Figure 3**). The Nup157 and Nup170 SeRP experiments did not reveal any enrichment (**Figure S6**) or onset curves (not shown) for the mRNAs in question, which is consistent with our RIP-qPCR analysis at the given threshold employed in the revised version of the manuscript (see response to point 2. and 1a. above).

4) *The results section of the text is challenging to follow. The presentation of the results is written in a branched, non-linear development of the scientific discoveries. An example for that is that in figure 2d and e the authors present results regarding the CTN subcomplex. The authors then move to a different subcomplexes in figure 2, different aspect of their work in figure 3, and then back to the CTN subcomplex in figure 4. In which, they describe the pre-existing steps for the results described in figure 2d and e. Readers which are not fluent in the structure of the nucleopore complex might find it confusing. I therefore suggest rethinking the order by which the results are presented.*

We agree that the data was challenging to present and we had discussed the best possible way of presentation prior to submission. We decided to present the data based on structural features and motifs rather than presenting one subcomplex after the other. We wanted to follow this logic: we first introduce the short linear motifs within with linker Nups that connect in between scaffolding components of primarily two subcomplexes, the inner ring and CF complex. Subsequently, we highlight the so-called insertion blades of Seh1 and Sec13. Both of these proteins moonlight on membrane coating complexes. Finally, we exploit the promiscuous coiled-coil interactions that compete between the CTN and CF subcomplexes.

We explain this rationale upfront in the results part: “We surveyed the known structural repertoire of nucleoporins for domains that could potentially engage in co-translational interactions because they (i) are small interaction motifs found in linker Nups; (ii) have obviously complemented the fold of another nucleoporin; or (iii) are shared between multiple complexes and thus could be promiscuous interactors (**Figure 1b**).”

Since the NPC architecture is formed by a complicated network of interactions that connects not only within but also across subcomplexes, the alternative representation the reviewer suggests would also not disentangle the data sets into subcomplexes. However, we made an effort to revise the text throughout for more clarity and better readability.

Minor concerns:

5) Line 143: “In order to elucidate the implementation of co-translation for faithful assembly of the NPC ...” please clarify the meaning of co-translation in this context. (e.g. “co-translational assembly”?)

We have revised this sentence as follows: “To elucidate how co-translational association of Nups contributes to faithful assembly of the NPC, we experimentally validated these motifs in a hypothesis-driven approach.”

6) Line 163: The RIP-qPCR data presented does not allow the strong conclusion at this point of the text.

We rephrased the respective sentence as follows: “Our RIP-qPCR data indicate a co-translational interaction of all three components of the CNT with the nascent chain of Nic96 (**Figure 1d**).”

7) Line 184: Nup49 should have been tested as well.

We want to point out that previous biochemical studies (Chug et al., 2015; Stuwe et al., 2015) highlighted the necessity of a the full trimer for association to Nic96. We therefore did not specifically target Nup49.

8) Line 233/Fig. S5c-f: Rather than showing individual, pre-selected SeRP profiles, IP/total RPKM enrichments for each mRNA should be presented in a scatter plot to get an overview of the nascent interaction landscape of the protein of interest.

We added volcano plots for all 10 SeRP experiments to **Supplementary Figure 6** as requested.

9) Line 259: The authors should support their interpretation that Sec13 may bind sec31 mRNA rather than the Sec31 nascent chain (NC), e.g. by SeRP. Sec31 is a rather large protein with a correspondingly long mRNA and presumably covered with many ribosomes. Since puromycin does not release NCs with 100% efficiency, and retaining a single sufficiently long nascent chain can result in the pull down of a given mRNA via NC-NC interactions, the enrichment using the RIP-qPCR approach may be decreased for long/highly translated mRNAs.

Our interpretation was based on the exceptionally strong enrichment of mRNA in the Sec31 RIP-qPCR experiment (**Figure 3a**, about 25-fold) and the significant puromycin dependent reduction of mRNA enrichment led to the conclusion that Sec31 may engage co-translationally with itself. We have done the respective SeRP experiment (**Figure 3b**), which underlines this conclusion. Sec13 which was in question before, was also shown to engage with nascent Sec31 in a co-translational manner at codon 437 (**Figure 3b**).

10) Fig. 4g: To validate the co-translational assembly pathway depicted (Gle2 → Nup116 → Nup82), Gle2 RIP-qPCR data testing for the enrichment of *nup82* mRNA should be added.

The respective data are presented in **Supplementary Figure 3c**. qPCR data of Gle2 supports that the dimer of Gle2 and Nup116 binds to Nup82 in a co-translational manner and cross validates findings observed in the RIP-qPCR experiment with Nup116 as presented in **Figure 1k** implying a linear assembly pathway which is seeded by Gle2 followed by the co-translational association of Nup116 and later Nup82.

11) Line 284/Fig. S6: Although generally conceivable, the interpretation that locally decreased translation efficiencies correlate with the assembly onsets as shown in Fig. S6 is questionable. Instead, the authors should test for an increased density of ribosomes before the onset of cotranslational assembly in the ribosome profiling data, indicating decreased local translation speed.

We have revised the respective and included the ribosome profiling data as requested. We have removed the respective sentence from the legend (now **Figure S8**).

12) Line 346: The authors probably meant to write that “Sec13 might rather associate with Sec31 post-translationally (after, not prior to) the co-translational entanglement of Sec31 with itself”

Yes, we have reinterpreted the data due to the new SeRP data regarding Sec13 and Sec31 (see above).

13) Line 375: The authors should provide data or clarify which of the data shown, and in which manner, support their conclusions about the stoichiometry during CTN formation and its assembly with Nic96.

Quantitative mass spectrometry was used to determine the relative enrichment of CTN and CF constituents relative to a Nsp1-StrepII stain (“wildtype”) which allows the determination of a ratio to a mutant strain. We included additional analysis and revised the respective plots in **Figure S9b**.

14) Fig. 5d: Indicate the significance of the differences between samples treated with CHX or puromycin to support the finding that alpha-beta deletion does not affect the cotranslational assembly of Nsp1 on nascent Nic96.

p-values are indicated in the revised **Figure 5d**. For the alpha-beta domain deletion p-values are not significant (*nup57*: 0.1011, *nic96*: 0.065; paired student’s t-test) upon puromycin-treatment with the current number of replicates. However, the puromycin dependent decrease of signal is clearly observed.

Reviewer #2

The following minor comments are suggestions that the authors may wish to consider including in the revision of their manuscript.

Minor points:

- 1) *The authors may want to explicitly explain why they compare cycloheximide stalled ribosomes with puromycin stalled ribosomes which they specify dissociated the nascent chain on line 157.*

Treating polysomes with cycloheximide or puromycin has become a state-of-the-art method when subjecting co-translational interactions as both compounds act translation specific (Duncan and Mata, 2011; Shiber et al., 2018). While the capturing of translating ribosomes with cycloheximide should maintain co-translational interactions and is particularly necessary for freezing ribosomes to the mRNA in ribosome profiling (positive control), the release of nascent chains with puromycin under high salt conditions was employed as a negative control (Blobel and Sabatini, 1971).

We revised the text as follows: *“This interaction was sensitive to the translation-specific inhibitor puromycin, which causes dissociation of the nascent chain and thereby enables the dissection of co-translational interaction from cryptic RNA binding activity (Blobel and Sabatini, 1971). Inverse tagging of Fas2 instead of Fas1 did not enrich for either mRNA, as expected (Figure S2c-S2e).”*

- 2) The authors suggest in line 306 “the sequences of the C-terminal coiled-coil segments of Nup57 and Nup82 align well with one another (Figure 5a)” yet reference a domain architecture. It may be more accurate to state the secondary structure or domain architecture aligns well.

We have rephrased the respective sentence as follows: *“The domain architecture of the C-terminal coiled-coil segments of Nup57 and Nup82 align well with each other (Figure 5a).”*

- 3) In the description of the Sec13 and Seh1 interactions with Nup145C and Nup85, respectively, the authors correctly cite Debler et al., Mol Cell for the Seh1-Nup85 interaction, but should also include the citation that originally reported the “domain invasion motif” in the Sec13-Nup145C complex 2 years earlier (Hsia KC, Stavropoulos P, Blobel G, Hoelz A. (2007). Architecture of a coat for the nuclear pore membrane. Cell 131, 1313-26). The citation should be inserted at the end of the following paragraph:

Inspired by the above findings, we wondered whether co-translational events may also specify assembly pathways for moonlighters that are members of multiple protein complexes. Two members of the Nup84-subcomplex, Sec13 and Seh1, are incomplete beta-propellers lacking one blade, that interact with the WD40 domain invasion motifs of Nup85 and Nup145C, respectively (Hsia et al. 2007; Debler et al., 2008).

We apologize for this oversight and have included the respective citation.

- 4) In the description of the Nup116 GLEBS interaction with Gle2 (page 7), the authors may want to consider citing the paper that reports the structure of the complex (Ren Y, Seo HS, Blobel G, Hoelz A. (2010). Structural and functional analysis of the interaction between the nucleoporin Nup98 and the mRNA export factor Rae1. PNAS 107,10406-11). The reference should be included after the following sentence:

The N-terminal GLEBS domain of Nup116 binds to Gle2 and is absent in Nup100 or Nup145N (Bailer et al., 1998; Ren et al., 2010) (Figure 2g).

We apologize for this oversight and have included the respective citation.

Reviewer 3:

1. Personally, I did not find Figure 1 or the first two paragraphs of the Results section (lines 105-136 particularly informative or helpful. These points seem almost trivial and could easily be summarized in the introduction, and Figure 1 does not present especially novel theoretical modeling. I am also not sure if I agree with the implication in Figure 1 that each step of an assembly pathway has the same probability of succeeding (or failing). Given that the introduction and discussion section already seem long, this portion of the texts seemed to only delay progression to the true results of the study. I recommend shortening or removing this section.

We agree that the assumption that each step has the same chance of success or failure is an over-simplification. It is more important to stress that linear pathways have a reduced number of steps which dramatically decreases the odds for overall success. We adjusted Figure and text accordingly (**Figure S1** and **Supplementary Text** of the revised version).

We still find it interesting and informative to point out that for very large assemblies, there is basically no chance for overall success if assembly pathways are not directed. We have expanded our model with explicit calculations to bring our case more clearly.

To nevertheless address the reviewer's point regarding readability, we have moved the kinetic model to the supplement and refer to it only very briefly in the main text.

2. Some parts of the text border on overinterpretation without parallel supporting evidence. In general, the data speak for themselves, and these statements are not necessary. Specific instances listed below:

Line 261-262: the implication of protein binding to the 3'UTR seems premature without further analysis, and because this is the only example of puromycin-independent interaction in the text.

This point has also been raised by reviewer 1. We agree and removed the respective statement as SeRP experiments highlighted the interaction of Sec13 with nascent Sec31.

Line 277-279: the statement "...the minimal requirement for co-translational interactions is the initial..." should be reworded or validated by chimeric experiments (e.g. transplanting the CCS1 to a different nascent protein to show sufficiency).

In the revised manuscript, we have obtained two novel assembly mutants in which coil-coiled segment 1 of Nup57 was successfully removed and substituted by the CCS1 of Nup82 either maintaining or removing the previously described alpha-beta domain of Nup57. Surprisingly, we find both mutants to be viable with a mild growth defect. Follow-up experiments addressing co-translational interactions show that the signal for *nup57* is depleted indicating that this helix specifies Nup57 for co-translational interactions. Consequently, this led to the reduction of co-purified Nup57 and Nup49 and enrichment of Nup82, but not Nup159 in a Nsp1-StrepII pull down (**Figures 5 and S9**).

Pages 9-10: the discussion of the two separate Nsp1 complexes heavily rely on previous studies, but the current study doesn't biochemically demonstrate that these distinct subcomplexes can be separated, and the presented data cannot directly support order of assembly. Language around these interpretations, e.g. line 302-303 "...tetramer forms co-translationally first and only subsequently binds to Nsp1" should be carefully curated.

We agree that RIP-qPCR does not resolve complexes in temporal resolution. However, conclusions that place co-translational interactions prior to post-translational interactions must be allowed. In example, if Nsp1 associates with Nup57 but not Nup49 co-translationally, it is obvious that the Nsp1:Nup57 heterodimer must form before Nup49 enters the complex (because *nup49* mRNA remains undetected). As suggested, we carefully revised the main text to make each single case in which we draw conclusions based on previously determined biochemical interactions more transparent.

Literature

- 1) Akey, C.W., Singh, D., Ouch, C., Echeverria, I., Nudelman, I., Varberg, J.M., Yu, Z., Fang, F., Shi, Y., Wang, J., et al. (2021). Comprehensive Structure and Functional Adaptations of the Yeast Nuclear Pore Complex. 1–28.
- 2) Allegretti, M., Zimmerli, C., Rantos, V., Wilfling, F., Ronchi, P., Fung, H., Lee, C.-W., Hagen, W., Turonova, B., Karius, K., et al. (2020). In cell architecture of the nuclear pore complex and snapshots of its turnover.
- 3) Bailer, S.M., Baldof, C., and Hurt, E. (2001). The Nsp1p Carboxy-Terminal Domain Is Organized into Functionally Distinct Coiled-Coil Regions Required for Assembly of Nucleoporin Subcomplexes and Nucleocytoplasmic Transport. *Mol. Cell. Biol.* *21*, 7944–7955.
- 4) Blobel, G., and Sabatini, D. (1971). Dissociation of mammalian polyribosomes into subunits by puromycin. *Proc. Natl. Acad. Sci. U. S. A.* *68*, 390–394.
- 5) Chug, H., Trakhanov, S., Hülsmann, B.B., Pleiner, T., and Görlich, D. (2015). Crystal structure of the metazoan Nup62•Nup58•Nup54 nucleoporin complex. *Science (80-)*. *350*, 106–110.
- 6) Duncan, C.D.S., and Mata, J. (2011). Widespread cotranslational formation of protein complexes. *PLoS Genet.* *7*.
- 7) Fischer, J., Teimer, R., Amlacher, S., Kunze, R., and Hurt, E. (2015). Linker Nups connect the nuclear pore complex inner ring with the outer ring and transport channel. *Nat. Struct. Mol. Biol.* *22*, 774–781.
- 8) Kim, S.J., Fernandez-Martinez, J., Nudelman, I., Shi, Y., Zhang, W., Raveh, B., Herricks, T., Slaughter, B.D., Hogan, J.A., Upla, P., et al. (2018). Integrative structure and functional anatomy of a nuclear pore complex. *Nature* *555*, 475–482.
- 9) Kosinski, J., Mosalaganti, S., Appen, A. Von, Teimer, R., Diguilio, A.L., Wan, W., Bui, K.H., Hagen, W.J.H., Briggs, J. a G., Glavy, J.S., et al. (2016). Pore Complex. *352*, 363–365.
- 10) Lin, D.H., Stuwe, T., Schilbach, S., Rundlet, E.J., Perriches, T., Mobbs, G., Fan, Y., Thierbach, K., Huber, F.M., Collins, L.N., et al. (2016). Architecture of the symmetric core of the nuclear pore. *Science (80-)*. *352*.
- 11) Melčák, I., Hoelz, A., and Blobel, G. (2007). Structure of Nup58/45 suggests flexible nuclear pore diameter by intermolecular sliding. *Science (80-)*. *315*, 1729–1732.
- 12) Onischenko, E., Stanton, L.H., Madrid, A.S., Kieselbach, T., and Weis, K. (2009). Role of the Ndcl interaction network in yeast nuclear pore complex assembly and maintenance. *J. Cell Biol.* *185*, 475–491.
- 13) Sharma, A., Solmaz, S.R., Blobel, G., and Melčák, I. (2015). Ordered regions of channel nucleoporins Nup62, Nup54, and Nup58 form dynamic complexes in solution. *J. Biol. Chem.* *290*, 18370–18378.
- 14) Shiber, A., Döring, K., Friedrich, U., Klann, K., Merker, D., Zedan, M., Tippmann, F., Kramer, G., and Bukau, B. (2018). Cotranslational assembly of protein complexes in eukaryotes revealed by ribosome profiling. *Nature* *561*, 268–272.
- 15) Solmaz, S.R., Chauhan, R., Blobel, G., and Melčák, I. (2011). Molecular architecture of the transport channel of the nuclear pore complex. *Cell* *147*, 590–602.
- 16) Stuwe, T., Bley, C.J., Thierbach, K., Petrovic, S., Schilbach, S., Mayo, D.J., Perriches, T., Rundlet, E.J., Jeon, Y.E., Collins, L.N., et al. (2015). Architecture of the fungal nuclear pore inner ring complex. *Science (80-)*. *350*, 56–64.
- 17) Ulrich, A., Partridge, J.R., and Schwartz, T.U. (2014). The stoichiometry of the nucleoporin 62 subcomplex of the nuclear pore in solution. *Mol. Biol. Cell* *25*, 1484–1492.
- 18) Yoshida, K., Seo, H.S., Debler, E.W., Blobel, G., and Hoelz, A. (2011). Structural and functional analysis of an essential nucleoporin heterotrimer on the cytoplasmic face of the nuclear pore complex. *Proc. Natl. Acad. Sci. U. S. A.* *108*, 16571–16576.

REVIEWERS' COMMENTS

Reviewer #1 (Remarks to the Author):

The revised manuscript has greatly improved and I generally support its publication. I nonetheless like to add a few comments and suggestions for the author's consideration.

- I prefer the title "Co-translational assembly orchestrates competing assembly pathways" as it better describes the findings of the manuscript.
- The authors may want to discuss why Nsp1 binds to the Nup82 CCS1 domain in the Nup57 CCS1 domain swap mutant background, but not the Nup57 CCS1 domain swap mutant itself.
- Line 273: I could not find clear evidence for an N-terminal bias of cotranslational interactions in the referenced publication (Shiber et al. 2018). Further, the existence of cotranslational assembly events involving the C-terminal half of a nascent chain does not disprove a putative general bias.
- Line 274: I agree that the finding on Nup85 is worth mentioning but overall, the data presented in Figure S8 does not support the general statement. Only Nup85 shows a 30 nt peak once the C-terminal oligomerization domain is exposed; Nup57 does not. Further, I advise against the use of the unintuitive "local translation efficiencies" and suggest using "local translation speed variations" or a similar term. I strongly prefer showing the ribosome profiling data and do not see the relevance of the "Translation efficiency profiles" provided in Figure S8.

Rebuttal Letter

We would like to thank the reviewer for the helpful critical comments and we have addressed them carefully in our final version of the manuscript.

Reviewer #1 (Remarks to the Author):

- I prefer the title “Co-translational assembly orchestrates competing assembly pathways” as it better describes the findings of the manuscript.

We have changed the title accordingly to “Co-translational assembly orchestrates competing biogenesis pathways.”

- The authors may want to discuss why Nsp1 binds to the Nup82 CCS1 domain in the Nup57 CCS1 domain swap mutant background, but not the Nup57 CCS1 domain swap mutant itself.

The weak onset in the *nup82*-ORF observed in the Nsp1-SeRP experiment under wildtype conditions (Fig. 5f) is C-terminally shifted towards CCS4 as compared to the *nup57*-ORF where interaction onset occurs at CCS1. Therefore, the motif contained in Nup57-chimeric mutants is likely not sufficient for co-translational binding of Nsp1. We added the following sentence to the main text:

“The reason why the Nup82 CCS1 is not sufficient for binding of Nsp1 when duplicated in the Nup57-ORF might be explained by the SeRP data that were obtained under wildtype conditions (**Fig. 5f**). It showed a slight enrichment that was C-terminally shifted with respect to *nup57*-ORF and located subsequent to the CCS3 of Nup82, which was not included in the Nup57 chimeric mutant. We thus speculate that the Nsp1 interaction with nascent Nup82 would normally be suppressed due to higher affinity of Nsp1 to Nup57 CCS1 but becomes relevant once effective binding to Nup57 is impaired (**Fig. 5g** and **Supplementary Fig. 8c**).”

Since we introduced the respective domain of Nup82 into Nup57 but not *vice versa*, Nup82 remained endogenous in all cases. The term ‘swap’ mutant was thus misleading and we replaced it with ‘chimeric’ mutant.

- Line 273: I could not find clear evidence for an N-terminal bias of cotranslational interactions in the referenced publication (Shiber et al. 2018). Further, the existence of cotranslational assembly events involving the C-terminal half of a nascent chain does not disprove a putative general bias.

We have revisited the Shiber et al. manuscript and agree that some, but not all of the investigated proteins showed an N-terminal onset (Fig 3a of Shiber et al.; <https://www.nature.com/articles/s41586-018-0462-y/figures/3>).

We therefore have removed the respective passage from the main text.

- Line 274: I agree that the finding on Nup85 is worth mentioning but overall, the data presented in Figure S8 does not support the general statement. Only Nup85 shows a 30 nt peak once the C-terminal oligomerization domain is exposed; Nup57 does not. Further, I advise against the use of the unintuitive “local translation efficiencies” and suggest using “local translation speed variations” or a similar term. I strongly prefer showing the ribosome profiling data and do not see the relevance of the “Translation efficiency profiles” provided in Figure S8.

We agree and have removed the respective figure that was only cited in this very particular context. This also becomes less relevant due to the above point of the reviewer.